

# MICS-Asia III: Multi-model comparison of reactive Nitrogen deposition over China

Baozhu Ge[1,2], Syuichi Itahashi[3], Keiichi Sato[4], Danhui Xu[1,5], Junhua Wang[1,5], Fan Fan[6], Qixin Tan[1,5], Joshua S. Fu[7], Xuemei Wang[8], Kazuyo Yamaji[9], Tatsuya Nagashima[10], Jie Li[1,2,5], Mizuo Kajino[11,12], Hong Liao[13], Meigen Zhang[1,2,5], Zhe Wang[1,2,14], Meng Li[15], Jung-Hun Woo[16], Jun-ichi Kurokawa[17], Yuepen Pan[1], Qizhong Wu[18], Xuejun Liu[19] and Zifa Wang[1,2,5]

[1] State Key Laboratory of Atmospheric Boundary Layer Physics and Atmospheric Chemistry (LAPC), Institute of Atmospheric Physics (IAP), Chinese Academy of Sciences (CAS), Beijing 100029, China

[2] Center for Excellence in Urban Atmospheric Environment, Institute of Urban Environment, Chinese Academy of Sciences (CAS), Xiamen 361021, China

[3] Environmental Science Research Laboratory, Central Research Institute of Electric Power Industry (CRIEPI), Abiko, Chiba 270–1194, Japan

[4] Asia Center for Air Pollution Research (ACAP), 1182 Sowa, Nishi-ku, Niigata, Niigata 950-2144, Japan

[5] Collage of Earth Science, University of Chinese Academy of Sciences, Beijing 100049, China

[6] Nanjing Intelligent Environmental Sci-Tech Co., Ltd., Nanjing, 211800, China

[7] Department of Civil and Environmental Engineering, University of Tennessee, Knoxville, TN 37996, USA

[8] Institute for Environmental and Climate Research, Jinan University, Guangzhou 510632, China

[9] Graduate School of Maritime Sciences, Kobe University, Kobe, Hyogo 658–0022, Japan

[10] National Institute for Environmental Studies (NIES), Tsukuba, Ibaraki 305–8506, Japan

[11] Meteorological Research Institute (MRI), Tsukuba, Ibaraki 305–8506, Japan

[12] Faculty of Life and Environmental Sciences, University of Tsukuba, Tsukuba, Ibaraki 305–8506, Japan

[13] School of Environmental Science and Engineering, Nanjing University of Information Science & Technology, Nanjing 210044, China

[14] Research Institute for Applied Mechanics (RIAM), Kyushu Univeristy, Kasuga, Fukuoka 816–8580, Japan

[15] Ministry of Education Key Laboratory for Earth System Modeling, Department of Earth System Science, Tsinghua University, Beijing 100084, China

[16] Division of Interdisciplinary Studies, Department of Advance Technology Fusion, Konkuk University, Seoul, 303-804, Korea

[17] Asia Center for Air Pollution Research (ACAP), 1182 Sowa, Nishi-ku, Niigata, Niigata 950-2144, Japan

[18] College of Global Change and Earth System Science, Beijing Normal University, Beijing 100875, China

[19] College of Resources & Environmental Sciences, China Agricultural University, Beijing 100193, China.

*Correspondence to:* Baozhu Ge (gebz@mail.iap.ac.cn)





**Abstract**: Atmospheric nitrogen deposition in China has attracted huge public attention in recently years due to the increasing anthropogenic emission of reactive nitrogen ($N_r$) and its impacts on the terrestrial and aquatic ecosystems. However, limited long-term and multi-site measurements have restrained the understanding on the mechanism of the $N_r$ deposition as well as the chemical transport model (CTM) improvement. In this study, the performance of the simulated wet and dry deposition for different $N_r$ species, i.e., particulate $NO_3^-$ and $NH_4^+$, gaseous $NO_x$, $HNO_3$ and $NH_3$, have been conducted using the framework of Model Inter-Comparison Study for Asia (MICS-Asia) phase III. Nine Models, including 5 WRF-CAMQ models, 2 self-developed regional models, a global model and a RAMS-CMAQ model, have been selected for the comparison. For wet depositions, observation data from 83 measurement sites of EANET, CREN, CAUDN, NADMN and DEE of China have been collected and normalized to compare with model results. In general, most models showed the consistent spatial and temporal variation of both oxidized N ($N_{ox}$) and reduced N ($N_{rd}$) wet depositions in China with the NME around at 50%, which is lower than the value of 70% based on EANET observation over Asia. Both the ratio of wet or dry deposition to the total inorganic N deposition (TIN) and the ratios of TIN to their emissions have shown the consistent results with the NNDMN estimations. The performance of ensemble results (ENM) was further assessed with the measurement from satellite. In different regions of China, the results showed that the simulated $N_{ox}$ wet deposition was overestimated in North East China (NE) but underestimated in south of China (SE+SW), while the $N_{rd}$ wet deposition was underpredicted in all regions by all models. The deposition of $N_{ox}$ have large uncertainties than the $N_{rd}$ especially in North China (NC), indicating chemical reaction process is one of the most importance factors that affecting the model performance. Compared to Critical load (CL) value, the $N_r$ deposition in NC, SE and SW reached or exceeded the reported CL value and exerted serious ecological impacts. The control of $N_{rd}$ in NC and SW and $N_{ox}$ in SE would be effective to mitigate the TIN deposition in these regions. More interestingly, the $N_r$ deposition in Tibet plateau with the high ratio of TIN/emission (~3.0), indicating a significant import from outside should be focused in the future due to its climatical influence to the sensitive ecosystem in whole China.

Keywords: Nitrogen deposition, multi-model comparison, China, reduced nitrogen, oxidized nitrogen



## 1 Introduction

Atmospheric Nitrogen (N) deposition is defined that N related gases and particles are deposited via precipitation (wet deposition) and not via precipitation (dry deposition) (Clark and Kremer, 2005). These depositions to the Earth's surface, are either close to the sources or in remote regions (e.g. chemical transformation and long-range transport of oxidized and reduced N, $N_{ox}$ and $N_{rd}$ hereafter), where is located far from human activities and labeled as the N-limited areas (Phoenix et al., 2006;Holtgrieve et al., 2011). Evidences show that the effects of reactive N ($N_r=N_{ox}+N_{rd}$) deposition to environment are numerous, including decreased biological diversity, increased soil acidification, and lake eutrophication (Clark and Tilman, 2008;Janssens et al., 2010;Holtgrieve et al., 2011;Phoenix et al., 2006;Galloway et al., 2004). Different human activities disturb the natural N cycle in serious manners (Galloway et al., 2004), for example using artificial fertilizer to increase crop production (Erisman et al., 2008) or relying too much fossil fuel for industrial production. The $N_r$ production increased from approximately 15 Tg N $yr^{-1}$ in 1860 to 187 Tg N $yr^{-1}$ in 2005 and more than 50% of them have been reported to deposit onto the ground (Nicolas and Galloway, 2008). In the past two decades, high rates of Nr deposition were widely documented in the developed countries such as America (Fenn et al., 1998) and Europe (Dise and Wright, 1995). Great efforts have been made to fight against these negative effects in USA to decrease $NO_x$, and the $N_{ox}$ deposition was decreased dramatically in recent years (Li et al., 2016). However, the growing human demand for food and energy at a global scale resulted in an increasing emission of $N_r$ into environment (Galloway et al., 2008), particularly in large developing countries like China and India (Chen et al., 2019a;Liu et al., 2013).

A nationwide estimation of long-term N deposition in China based on the bulk measurements as well as summarizing from reported references in 270 sites by Liu et al. (2013) shown an increasing rate of 0.41 kg N $ha^{-1}$ per year from 1980 to 2010. Different from the increasing importance of $N_{rd}$ deposition due to apparently decreasing of $N_{ox}$ that benefit from the air quality control in USA in past decades (Li et al., 2016), the ratio of $N_{rd}/N_{ox}$ recorded from bulk/wet deposition decreased from 5 in 1980 to 2 in 2010 indicating the more and more important role of $N_{ox}$ in China (Liu et al., 2013). The ratio in highly developed regions such as North China Plain was even lower than 1 in recent years (Pan et al., 2012). However, very limited long-term observations in China challenge our capacity to understand and control the increase of $N_r$ deposition. The published long-term N deposition monitoring network, which includes the Acid Deposition Monitoring Network in East Asia (EANET, http://www.eanet.asia/index.html), the National wide Nitrogen Deposition Monitoring Network (NNDMN) established since 2010 by China Agriculture University (CAU) (Xu et al., 2015), and the Chinese Ecosystem Research Network (CERN) in North China Plain established by Chinese Academy of Science (Pan et al., 2012), the Acid Rain Monitoring Network run by the China Meteorological Administration (CMA-ARMN) (Tang et al., 2007;Tang et al., 2010;Ge et al., 2011), and the National Acid Deposition Monitoring Network (NADMN) (Li et al., 2019b), have been identified with many shortcomings, e.g. the scattered monitoring sites as well as the



uncompleted recorded data, due to high cost of the measurement against the unstable
financial support. Chemical Transport Model (CTM) simulation is another option to
offset these drawbacks and also to quantify long-range transport of deposition in a
global or regional map (Seinfeld and Pandis, 2006). It is important to know the
accuracy of the CTM before it is employed to investigate the spatial and temporal
variation of the depositions. Hayami et al. (2008) and Mann et al. (2014) referred that
different parameterization set in CTMs may result in large variations, and the
multi-model ensemble mean (ENM) shows better performance than most single one
(Carmichael et al., 2002;Hayami et al., 2008;Holloway et al., 2008;Wang et al., 2008).
Besides, to better localize applications of CTM, the comprehensive evaluations of the
strengths and weaknesses of current CTMs for simulating the acid deposition as well
as their precursors in a unified framework, with certain regulated rules and same
inputs to models, should be more critical and effective.
Model Inter-Comparison Study for Asia (MICS-Asia) gives an opportunity to
investigate the CTMs application with different models in Asia. MICS-Asia was
initiated in 1998 with the target of long-range transport and deposition in $SO_4^{2-}$ in the
first stage (MICS-Asia phase I) (Carmichael et al., 2002) and sulfur, nitrogen and
ozone in the second stage (MICS-Asia phase II) (Carmichael et al., 2008). The
findings concluded and the methodologies developed in the previous inter-comparison
studies undoubtedly contributed to common understandings of the performance and
uncertainties of CTMs applications in East Asian (Hayami et al., 2008;Carmichael et
al., 2008;Han et al., 2008;Wang et al., 2008). The comprehensive multi-model
inter-comparison study on the acid deposition in China is becoming urgent issue as
the high emissions in China causing acid deposition in neighboring countries (Lin et
al., 2008;Kajino et al., 2011;2013;Itahashi et al., 2018),. In this study, one year
simulated $N_r$ depositions, i.e. $N_{ox}$ and $N_{rd}$ in both wet and dry deposited ways to the
surface ground, using the framework of MICS-Asia III (MICS-Asia phase III), have
been compared with each other and validated by the observed wet deposition from
EANET, NNDMN, CREN and the Department of ecological environment (DEE,
former named as Environmental Protect Administration, EPA) over the whole China.
The rationale of the performance of the ENM results were also discussed by
comparing with the Vertical Column Density (VCD) from satellite and the emission
inventories. Finally, the uncertainties of the pathways to $N_r$ depositions as well as its
ecological impacts have been quantified. The results from this study will not only
provide important reference for establishing a suitable N deposition model, the
localized application of CTMs in China will also be tested.
**2 Framework of intercomparison in MICS-Asia III**
**2.1 Description of the participant models**
In the phase III of MICS-Asia, a total of 14 chemical transport models (CTM,
M1-M14) participated in the topic of comparison and evaluation of current
multi-scale air quality models (Named as topic 1 in MICS-Asia III). The same number
index has been used in the deposition comparison part defined in aerosol and ozone
comparison reported by Chen et al. (2019b) and Li et al. (2019a). However, the fully
coupled online Weather Research and Forecasting model with chemistry



(WRF-Chem), which has been indexed as M7-M10, was not included in the
deposition comparison part in the overview of model inter-comparison and evaluation
for acid deposition in Asia (Itahashi et al., 2019). Briefly, Weather Research and
Forecasting model coupled with Community Multi-scale Air Quality (WRF-CMAQ)
has been numbered as M1-M6, with different version of v5.0.2 for M1 and M2, v5.0.1
for M3 and v4.7.1 for M4-M6. M11 and M12 were the independent models developed
by Japan and China, named as NHM-Chem (Kajino et al., 2019) and the nested air
quality prediction model system (NAQPMS), respectively. A global three-dimensional
chemical transport model (GEOS-Chem v9.1.3), numbered as M13, was also used as
the long-range transport and future change prediction in MICS-Asia III. The last
number of M14 was represented as the Regional Atmospheric Modeling System
coupled with CMAQ (RAMS-CMAQ). It should be noted that the last two models,
i.e., M13 and M14, were not driving by the "standard" meteorological fields from
WRF v3.4.1 model. Basic information about the configuration of each model was
summarized in Table 1. More detailed description could also be found in previous
studies (Itahashi et al., 2019;Chen et al., 2019b;Li et al., 2019a).
**2.2 Model inputs and simulation domain**
As that mentioned by Chen et al. (2019b), same ("standard") meteorological fields,
emission inventories and boundary conditions have been prepared for the CTMs
inter-comparison in MICS-Asia III to reduce the uncertainties from model inputs.
However, some models such as M13 and M14 were imported "non-standard" inputs
due to their specific characteristics. The "standard" meteorological inputs were
simulated by WRF v3.4.1 with the initial and lateral boundary conditions from the
National Centers for Environmental Prediction (NCEP) Final Analysis (FNL) data.
Four dimensional data assimilation (FDDA) nudgings have been adopted every 6
hours to improve the accuracy of the meteorological parameters simulation. The
assimilated meteorological fields from the Goddard Earth Observing System 5
(GEOS5) of the US 25 National Aeronautics and Space Administration (NASA)
(https://gmao.gsfc.nasa.gov) were used to drive M13. The M14 model was driven by
RAMS with the same FNL data for nudging as the "standard" WRF simulation, which
is developed by Pielke et al. (1992). For the emission inputs, all the participant model
were using the same emission inventory, which included the MIX anthropogenic
emissions over Asia developed for MICS-Asia Phase III (Li et al., 2017), the biogenic
emissions calculated by the Model of Emissions of Gases and Aerosols from Nature
(MEGAN) version 2.04 (Guenther et al., 2006), and the biomass burning emissions
from Global Fire Emission Database (GFED) version 3 (van der Werf et al., 2010).
Besides, the $SO_2$ emissions from volcano were collected from AEROCOM program
(https://aerocom.met.no/ DATA/download/emissions/AEROCOM_HC/volc, last ac-
cess: 11 September 2019, Diehl et al., 2012; Stuefer et al., 2013). MICS-Asia Phase
III provided two sets of lateral boundary conditions derived from GEOS-Chem (Bey
et al., 2001) and CHASER (Sudo et al., 2002), respectively. The boundary conditions
from GEOS-Chem were run with 2.5 °×2 ° resolution and 47 vertical layers, while
those from CHASER were run with 2.8 °×2.8 ° and 32 vertical layers. M4, M5, M6,
M11 and M12 used the output from CHASER as the boundary conditions, and M1,


M13 and M14 were from GEOS-Chem. Only M2 used the default boundary condition
field provided in CMAQ.
The "standard" simulation domain covers the region of East Asia (15.4°S-58.3 °N,
48.5 °-160.2 °E) with 180×170 grids at 45 km horizontal resolution. M1-M6, M11 and
M12 followed "standard" simulation domain, while M13 and M14 employed different
modeling domains with 0.5 ° latitude × 0.667 ° longitude and 64 × 64 km, respectively.
In this study, the analyzed region was only focused in China and all participant
models covered it. Therefore, simulated reactive N depositions in each model can be
analyzed and compared to show the performance of the participant models. All
models output of N depositions have been classified as oxidized N ($N_{ox}$ = gHNO$_3$ +
gNO$_x$ + pNO$_3^-$, including gaseous nitrate acid, NO$_x$ and particulate nitrate) and
reduced N ($N_{rd}$ = gNH$_3$ + pNH$_4^+$, including gaseous ammonia and particulate
ammonium) for comparison.
**2.3 Observation data**
China has large area with almost 5,500 km from south to north (3.5 °-53.3 °N) and
5,200 km from west to east (75.5 °-135 °E), which go through the coastal to inland and
through tropical to Frigid Zone. Only 8 sites located in Guangdong, Fujian, Sichuan
and Shanxi in EANET were not sufficient to show the real performance of CTMs in
China. Besides the 8 EANET sites, 83 sites in total with daily or weekly and even
yearly routine recorded data from the CERN (Pan et al., 2012), NNDMN (Xu et al.,
2015;Liu et al., 2013) and DEE of Guangdong, Liaoning and Xinjiang province as
well as Shanghai have been employed in this study to compare the simulated wet
deposition in MICS-Asia III in China. Figure 1 displayed the location of 83
measurement sites as well as the divided regions of the whole China. There were 50
urban sites and 33 rural sites. More detailed information of each measurement site
could also be found in Table S1 in the supplementary documents.
The daily wet deposition was measured by wet-only sampler to collect precipitation
samples during the rainfall event in EANET. Analysis methods for NO$_3^-$ and NH$_4^+$
were based on ion chromatography and checked by ion balance and conductivity
agreement. Detailed description could be found on manual (EANET, 2010). Daily
rainwater samples at 10 sites located in North China Plain were collected using a
custom wet-dry automatic collector (APS-2B, Xianglan Scientific Instruments Co.,
Ltd., Changsha, China) in CREN. Inorganic N, including NO$_3^-$ and NH$_4^+$, in the
precipitation samples was determined using an ion chromatography system (Model
ICS- 90, Dionex Corporation, Sunnyvale, CA, USA) and the standard laboratory
procedure of LAPC (Wang et al., 2012). The detection limit (DL) of N for this
instrument was 5 μg l$^{-1}$. Detailed description could be found in the research of Pan et
al. (2012). The wet/bulk NO$_3^-$ and NH$_4^+$ deposition data from 25 sites of China
Agricultural University Deposition Network (CAUDN), which was renamed as
NNDMN in China since 2010, have been collected and reanalyzed as yearly data (Xu
et al., 2015;Liu et al., 2013). At all monitoring sites precipitation samples were
collected using precipitation gauges (SDM6, Tianjin Weather Equipment Inc., China)
located beside the DELTA systems (ca. 2m, DEnuder for Long-Term Atmospheric
sampling). After collecting, the samples have been analyzed in CAU's laboratory



based on the standard laboratory procedure of CAU (Xu et al., 2015). Routine $NO_3^-$
and $NH_4^+$ wet depositions collected in each rainfall event at 40 sites have been
provided by the DEE of Guangdong, Liaoning and Xinjiang province as well as
Shanghai city. The analyzed procedure was followed as the laboratory procedure of
China National Environmental Monitoring Centre (CNEMC).
All data from daily or rainfall event collecting samples at each type of
measurement sites has been summarized and normalized as monthly wet deposition
data to compare with the monthly simulation in MICS-Asia III in this study, except
the yearly data provided by NNDMN. VCD of $NO_2$ from SCIAMACHY
(http://www.temis.nl/airpollution/no2col)      and      $NH_3$      from      IASI
(http://ether.ipsl.jussieu.fr/ether/pubipsl/iasial2/iasi_nh3) have also been used to
compare with the total deposition as well as the emissions.
**3 Results**
**3.1 Validation of wet deposition**
**3.1.1 Yearly comparison and monthly variation of oxidized N**
Yearly simulated wet deposition of $N_{ox}$ has been evaluated by observed nitrate wet
deposition in 83 sites over China. Table 2 listed the statistical parameters of simulated
wet deposition of $N_{ox}$ compared with the observed data in rural and urban sites of
China. In all sites, M1, M5 and M11 overestimated the yearly wet deposition of $N_{ox}$
with Normalized Mean Bias (NMB) of +30.3%, +55.4% and +67.2%, respectively.
M6, M12 and M13 simulated almost comparable results with NMB of -6.8%, +0.6%
and +0.1%, respectively. M2, M4 and M14 underestimated the yearly wet deposition
of $N_{ox}$ with NMB of -38.7%, -10.7% and -47.4%, respectively. The NME was almost
around at 50% with highest 82.2% in M11, which is lower than EANET observation
over Asia with the value at 70% by Itahashi et al. (2019). However, the correlation
coefficients R was around 0.2~0.3 (n=83) which is lower than EANET data (0.3~0.4,
n=54) (Itahashi et al., 2019). In order to eliminate influences from rainfall
uncertainties (R=0.82), the volume weighted mean (VWM) concentration of $N_{ox}$ in
precipitation has also been evaluated. In contrast to the low R value of yearly wet
deposition of $N_{ox}$, the correlation R increased to almost 0.5 for the VWM
concentrations. Approximately 50% of model results were corresponded within the
percentages within a factor of 2 (FAC2). M1 and M13 performed better agreement
with 60% and 59% within FAC2, while M2 and M14 showed only 36% and 45%
agreement within FAC2. All of ground surface measurement sites have been divided
into 49 urban sites and 34 rural sites according to their location. Overall, all the
models showed better performance with the R in 0.2~0.4 and FAC2 in 50%~60% in
urban sites than that of R in 0.05~0.3 and FAC2 in 40%~50% in rural sites. This
difference may not due to the uncertainties in rainfall simulation, as the simulated
VWM concentration of $N_{ox}$ in precipitation may eliminate the rainfall uncertainties,
and also shows better agreement in urban than that in rural sites (Table 2).
Figure 2 showed the percentile box plot the yearly wet deposition of $N_{ox}$ simulated
by 9 participant models in five regions of China (i.e., North China (NC), Northeastern
China (NE), southeastern China (SE), northwestern China (NW), southwestern China
and Tibet Plateau (SW+TP)). Site by site validation of both the yearly wet deposition





and VWM concentration of $N_{ox}$ simulated by each model were also displayed in
Figure S1. The model results showed large intra-region or inter-region uncertainties,
especially in NC, NE and SE. The highest wet deposition of $N_{ox}$ simulated by M11
was almost 3~4 times of the lowest result simulated by M14 in the above regions
(Figure 2). Specifically, two models simulated 30-50% higher of $N_{ox}$ wet deposition,
while four models were 20~40% lower compared to the averaged observations in NC
with the averaged value 6.5 kg N $ha^{-1}$ $a^{-1}$. For the wet deposition of $N_{ox}$ in SE and
SW+TP, most of the participant models were more than 50% underestimated with the
largest underestimation of 75% from M14, even though the precipitation in this region
was overestimated. Besides, the divergence of observed $N_{ox}$ wet deposition between
different sites in NC, SE and SW, which was shown as the length of the red box in
Figure 2a, 2d and 2e, was significantly larger than the multi-models results. The
scattered distribution of the measurement sites in these regions was responsible for the
large divergence in observations. However, most of the participant models failed to
capture the large difference, indicating that the coarse grid in MICS-Asia III (45 km)
was not suitable for the performance of detailed characterization at a local scale. A
global assessment of the ensemble simulated wet depositions in the Task Force on
Hemispheric Transport of Atmospheric Pollutants (TF HTAP) by Vet et al. (2014) also
indicated the underprediction of the models in a number of sites in north America,
Europe, Central Africa and part of East Asia. The underprediction in Europe was
found due to the large underpredictions of precipitation depth, while the reason for
East Asia is still unknown. However, most of the models overestimated the wet
deposition of $N_{ox}$ in NE. Several models including M1, M5 and M11 simulated more
than 10 kg N $ha^{-1}$ $a^{-1}$ $N_{ox}$ wet deposition, almost double higher than the observed value
of 5 kg N $ha^{-1}$ $a^{-1}$. Both the multi-models and the observations showed very low
values of 3-4 kg N $ha^{-1}$ $a^{-1}$ $N_{ox}$ wet deposition in NW, where the precipitation depth
was very low compared to the other regions of China (Figure S1).
Further evaluations in temporal variations both in urban and rural sites of NC and
NE have been displayed in Figure 3. Generally, all of the models and observations
performed high level of depositions in spring and summer and low value in winter in
the two regions. High depositions were due to large precipitation depth in rainy
season. However, this was not always true in some urban sites. For example, higher
depositions of $N_{ox}$ were observed in May and June with lower rainfall volume than in
July and August with higher rainfall in the urban sites of NC. Similar cases were
found at urban sites in NE. Previous studies confirmed there is a decreasing trend in
the variations of chemical components in precipitation as the rainfall evolution
(Aikawa and Hiraki, 2009;Aikawa et al., 2014;Xu et al., 2017). If the rainfall lasted
long enough, or rainfall volume was large enough, the concentrations of chemical
components in precipitation remained at low levels and were attributed to the effects
of the in-cloud scavenging process. That is, the large rainfall in an event may not
cause the high level of monthly wet depositions due to the low level of in-cloud
deposition compared to the wet deposited by several different precipitation events,
especially in polluted regions in urban sites. Unfortunately, only monthly data of wet
deposition as well as precipitation have been compared in this MICS-Asia III.



Detailed comparison with the rainfall event should be considered in the future.
**3.1.2 Yearly comparison and monthly variation of reduced N**
Simulated wet deposition of $N_{rd}$ in MICS-Asia III has been evaluated using the
multi-source of observations from the same sites as referred in $N_{ox}$. It is shown in
Table 3 that all of the models underestimated the $N_{rd}$ wet depositions with the
negative NMB both in urban and rural sites. Although little difference between rural
and urban sites was found in M11 and M14, a better performance in rural area was
manifested from the lower NMB and higher FAC2 in rural sites than the urban sites in
most of models (-13.6%~-23.2% vs -37.3%~-45.6% for NMB and 55.9-70.6% vs
42.9-55.1% for FAC2, except M11 and M14). The underestimation of the simulated
$N_{rd}$ wet depositions was also found in the VWM concentration of $N_{rd}$ in precipitation
with similar NMB and FAC2. However, compared with the wet deposition, the
correlation between the simulated and observed $N_{rd}$ VWM concentration in
precipitation was significant with the R increased from ~0.3 to ~0.8, which was
similar with that shown in $N_{ox}$. This indicated the simulated VWM concentration of
$N_{rd}$ in precipitation by MICS-Asia III has better performance in spatial variation than
the simulation of $N_{rd}$ wet deposition over China.
Specifically, the performance of $N_{rd}$ wet deposition prediction in MICS-Asia III has
also been validated in five regions through the percentile box plot in Figure 4. Site by
site validation of both the yearly wet deposition and VWM concentration of $N_{rd}$
simulated by each model were also displayed in Figure S2. Different from that found
in $N_{ox}$, almost similar behavior prediction has been found in same models, i.e.,
CMAQ models in M1~M6 but except M12 which was driven by different
meteorological model. Other regional model as well as global model showed
significantly different percentile distribution in all regions. Overall, both the medium
and mean value of $N_{rd}$ wet deposition were underestimated apparently in NC, SE and
SW+TP, while similarly in NE and NW. The underestimation in NC was largely due
to the under prediction in summer time not only in urban sites (Figure 3e) but also in
rural sites (Figure 3f). Unfortunately, we cannot obtain the convincing temporal
variation in SE and SW since the scarcely monthly data in these two regions (only one
or two sites in each region). In NE, most of the models predicted similar temporal
variations of $N_{rd}$ wet deposition, especially the high depositions in summer time.
**3.2 Map of wet deposition among participant models**
**3.2.1 Wet deposition of oxidized N**
Figure 5 showed the map distribution of the yearly $N_{ox}$ wet deposition simulated by
each participant model, the ENM results and the observed results over China. Most
models performed the similar spatial pattern with high level of deposition in central to
eastern China and low level in western China. However, the threshold value in the
Hotspot areas (from light yellow color to orange and red colors) varied significantly
among the models and the average is much higher than the Nr deposition threshold
value of 10 kg N ha$^{-1}$ to the temperate ecosystems suggested by Bleeker et al.(2011).
For example, M1, M5 and M11 simulated very high wet deposition of $N_{ox}$ (almost
reach at 20 kg N ha$^{-1}$) in the middle Yangze River and Yangze River Delta (YRD),
basin of Sichuan Province, south of NC and Liaoning Province located in NE. In


contrast, M2 and M14 failed to perform the relative hotspot $N_{wox}$ in such areas, and
M4, M6, M12 and M13 showed the obscure hotspot with small value of 10 kg N ha$^{-1}$.
The significant differences do not only exist between different models but also in the
same model CMAQ, i.e., M1, M2, M4, M5 and M6. Since most models were driven
by the meteoroidal field and standard emission input except M13 (Geos-Chem) and
M14 (RAMS-CMAQ), the differences in simulated $N_{ox}$ wet deposition should come
from the CTMs themselves, such as the diffusion and convection process, the
oxidation and chemical transformation as well as the wet scavenging and deposition
processes. The comparison of the long lifetime specie CO (Kong et al., 2019) and
weak chemical activity specie BC (Chen et al., 2019b) revealed that the model
uncertainties are less than other species, i.e., $O_3$ (Li et al., 2019) and $NO_3^-$ (Chen et al.,
2019b) which are strong chemical activity and short lifetime in the atmosphere. These
results indicated that the chemical reaction process rather than the diffusion and
convection process is one of the most important factors affecting the model
uncertainties in MICS-Asia III.

**3.2.2 Wet deposition of reduced N**

Figure 6 showed the map distribution of the reduced N ($N_{rd}$) wet deposition over
China. All of the models performed similar spatial pattern with high values in central
and eastern China but low level of deposition in NW and northwestern of NE.
Compared with the $N_{ox}$, little differences of the simulated $N_{rd}$ wet deposition were
found among 9 models except M11, which predicted significant lower values. $N_{ox}$ wet
deposition of five agricultural dominant provinces including Shandong, Henan, Hubei,
Hunan and Anhui is higher than the threshold value of 10 kg N ha$^{-1}$ according to the
simulated results by most models. Unfortunately, little observations in these areas
made it harder to validate their truthfulness. Evidence showed the high level of $N_{rd}$
wet deposition over the threshold based on the observations in Hebei, YRD and Pearl
River delta (PRD). Almost all of the models were under predicted in these areas. Liu
et al. (2013) reported the important contribution of $N_{rd}$ to the total N deposition in
China based on the long-term national scale of observed nitrogen deposition data. In
the agricultural predominant areas, the ammonia emission is the main contribution to
the $N_{rd}$ deposition (Liu et al., 2011 AE review; Kang et al., 2016). Thus, although the
rarely observation cannot support the simulated high level of $N_{rd}$ wet deposition in
agricultural predominant regions, i.e., Shandong, Henan, Hubei, Hunan and Anhui,
where the simulated $N_{rd}$ wet depositions are higher than 10 kg N ha$^{-1}$, it may be more
convincing that the $N_{rd}$ wet deposition is higher than the threshold value according to
the confirmed underestimation both in agricultural areas (i.e., Hebei) and in
non-agricultural areas (i.e., YRD, PRD).

**3.3 Comparisons among participant models for reactive N depositions**

**3.3.1 Coefficient of variations for N depositions in MICS-Asia III**

Besides the wet deposition of oxidized and reduced N, dry deposition was also an
important process for the total deposition part in China (Liu et al., 2013;Pan et al.,
2012). Coefficient of Variation (hereinafter, CV), defined as the standard deviation
divided by mean value of all selected model results, with large value denoting lower
consistency among the models, is applied for model comparison of simulated reactive





N depositions both for dry and wet deposition process in MICS-Asia III. Figure 7
shows the distribution of CV for each type of simulated reactive N depositions. Since
the low level of mean values of deposition are more likely to associate with higher CV,
the gridded CV was only calculated in the area with the simulated depositions higher
than 0.5 kg N ha$^{-1}$ (hereafter, analyzed value) in this study. As it is shown in Figure 7,
the spatial distribution of CV only covered Eastern China, Southern China and
Northeast China, which indicated that the quarterly and yearly fluxes of reactive N
deposition in these regions was higher than the analyzed value. For annual case, the
CV value of $N_{rd}$ was lower compared with $N_{ox}$ both for dry and wet depositions. This
means the multi-model simulations were more consistent in $N_{rd}$ depositions than in
$N_{ox}$ depositions. More specifically, the $N_{rd}$ in wet deposition has lowest CV values
followed by $N_{rd}$ in dry deposition and then the $N_{ox}$ in wet and dry deposition, which
suggested the simulated wet deposition of $N_{rd}$ had less uncertainties than the other
types of reactive N depositions.
More complicated patterns were shown in seasonal variations of each type of
deposition. The simulated $N_{ox}$ for dry deposition in Figure 7 (a) showed larger
uncertainties in southern China (south of 30 °N, with the CV > 0.4) than that in
northern China (north of 30 °N, with the CV <0.3) in all seasons except summer.
Similar spatial and temporal patterns of the CV values were found in $N_{rd}$ dry
deposition. It is worth noting that the large CV values with the range of 0.4-0.6 were
exhibited in Central China (i.e., Henan, Hebei and Shandong provinces) during
summer and autumn in spite of the high flux of $N_{rd}$ dry deposition in these regions
(map distribution of $N_{ox}$ and $N_{rd}$ dry deposition simulated in 9 participant models was
displayed in Figure S3 and Figure S4 of the supplementary documents). This
delivered an important message that the uncertainties of the physical and chemical
processes in the participant models, including gas-particle equilibrium (Ge et al.,
2019), dry deposition parameter scheme (Zhang et al., 2003), transportation as well as
the chemical reaction with other acidifying substances (Liu et al., 2019), in the
regions of high emission originated from agricultural activities in growing seasons
may lead to significant deviation of simulated $N_{rd}$ dry depositions.
For wet deposition of $N_{ox}$, large uncertainties were located in southern China in
summer and autumn with the CV values higher than 0.6 compared with the CV values
lower than 0.4 in other regions (Figure 7c). Anyway, this high value of CV was not
found in the summer time of simulated $N_{rd}$ wet deposition (Figure 7d). Due to the
high portion of summer time flux to the total annual wet deposition, high CV value in
$N_{ox}$ contributed to the most important part of the significantly larger annual CV value
than that shown in the $N_{rd}$ case. Due to the same rainfall input for the wet deposition
in the framework of MICS-Asia III except model 13 and 14, the different CV values
for $N_{rd}$ and $N_{ox}$ in same region (i.e., lower CV values of $N_{rd}$ wet depositions in NC, SE
and Central China) would attribute to their precursors concentration in the air mass as
well as the different wet scavenging processes (Seinfeld and Pandis, 2006). This has
been discussed in the following section.
**3.3.2 Comparison of precursors in the air mass**
As we all know, depositions both from dry and wet part of a certain substance were



originated from its precursor in the air mass. The uncertainties of the nitrogen related
species in the air mass simulated during MICS-Asia III were therefore an important
index for estimating the performance of deposition simulations. Figure 8 showed the
distribution of CV for $NO_x$, particulate $NO_3^-$, gaseous $NH_3$ and particulate $NH_4^+$ in the
air mass simulated by the 9 participant models during four seasons as well as the
annual mean values. There were significant seasonal variations among the spatial
patterns of the CV for each type of the N related air pollutants. It is interesting to note
that not only the seasonal variations but also the spatial patterns of the simulated
precursors' CV were reasonably consistent with those previously shown in the
deposition part (Figure 7). For example, high CV values were found in the simulation
of particulate $NO_3^-$ in Southern China during summer, reaching to or even higher than
0.8 in SE China (Figure 8b). The high CV values were also found in the summertime
of $N_{ox}$ wet deposition (Figure 7c). As the most important precursor of $N_{ox}$ wet
deposition (Pan et al., 2012), the correlated consistence between the precursor and the
deposition is reasonable. This has also been proved in the distribution of CV values in
$NO_x$ (Figure 8a) and $N_{ox}$ dry depositions (Figure 7a) during autumn and winter.
However, only uncertainties in precursors cannot explain everything, for example, the
high CV values of $N_{ox}$ wet deposition in south China was corresponding to the low
CV values of $NO_3^-$ in autumn. Some other factors, such as the scavenging process
might be responsible for the unknown-uncertainties. Xu et al. (2017;2019) first
compared the below-cloud wet scavenging coefficients based on the different
estimation methods and found the magnitude difference between each type of
methods. Thus, more detailed comparison such as in-cloud and below-cloud wet
scavenging coefficients in each participant model should be carried out in the next
phase of MICS-Asia.
As the most important precursor of $N_{rd}$ dry deposition, gaseous $NH_3$ also showed
large CV values in central China during summer time (> 0.6). There were also
significant high CV values in south of Yangtze River during autumn and winter period
(0.7-0.8 in south of Yangtze River vs 0.3-0.5 in north of Yangtze River). The similar
pattern but not significant uncertainty was found in the simulated $N_{rd}$ dry deposition
(0.3-0.4 vs 0.2-0.3 in Figure 7b). Different from the particulate $NO_3^-$, very low CV
values were shown in particulate $NH_4^+$ during summer leading to the less deviation of
simulated $N_{rd}$ wet deposition than the $N_{ox}$. Therefore, the performance of the
precursors' simulation was highly correlated with their depositions, while other
factors such as wet scavenging process might lead to the unknown uncertainties but
this need to be verified in the future.
**4 Discussion**
**4.1 Ensemble results of reactive N deposition and comparison with satellite**
Wang et al. (2008) first presented the ENM depositions of acidify species over East
Asia based on MICS-Asia II simulations and found that the ENM afford better skill in
simulating wet depositions than each single model. In the phase III of MICS-Asia, the
ENM value of wet depositions both for $N_{ox}$ and $N_{rd}$ has also been validated by
observations and shown in Figure 5l and Figure 6l. The simulated $N_{ox}$ wet deposition
and VWA concentration in rainfall exhibited larger dispersion around 1:1 line with the





correlation coefficients R were 0.23 and 0.54 in 83 sites over China than that found in
$N_{rd}$, which is concentrated around 1:2 line with the correlation coefficients R were
0.32 and 0.8. This implicated the ensemble-mean value of simulated $N_{ox}$ wet
deposition has large uncertainties, while $N_{rd}$ wet deposition was under predicted by a
factor of two in MICS-Asia III. Compared to each single model, the ensemble-mean
showed higher R value than most of single models. However, due to lack of direct
observation of dry deposition, the validation for dry and total deposition of reactive N
cannot be achieved. Instead, the column densities from satellite and emissions spatial
distribution were employed to address the reasonability of the ensemble-mean of four
types of reactive N depositions simulated in nine models. As displayed in Figure 9,
dry depositions of $N_{ox}$ and $N_{rd}$ were concentrated in NC, YRD and Henan province,
which is correspondence to the distribution of their emissions and VCDs, respectively.
Meanwhile, wet depositions of $N_{ox}$ and $N_{rd}$ were centered at central China such as
Hubei and Hunan province as well as Chengyu regions. Especially, there were high
wet depositions of $N_{rd}$ in south west of Hubei province and north east of Chengdu city,
where high values of emissions and the VCDs for $NH_3$ were absent. These regions
loading with high wet depositions were mainly due to the high volume of rainfall (for
more details at Figure S5) and the long-range transport of acidic substances (Ge et al.,
2011).
Another interesting phenomenon was the different allocation of high values
between depositions and VCD for $N_{ox}$ and $N_{rd}$. For example, low depositions were
loading in East China with high value of VCD for $N_{ox}$ as the left panel of Figure 9
shown. While as it was displayed in the right panel of Figure 9, large depositions of
$N_{rd}$ in East China were corresponding to low level of VCD on the contrary. The whole
emissions of $NO_x$ and $NH_3$ were similar at 8 kg N•ha$^{-1}$ (8.3 kg N•ha$^{-1}$ and 8.7 kg
N•ha$^{-1}$ for $NO_x$ and $NH_3$, respectively) in China statistically from MICS-Asia III
emission inventory. This implicated the allocation of both $NO_x$ and $NH_3$ between the
deposition to the surface ground and staying in the atmosphere were conserved to
their emission into the air. This conservation data from different sources also
implicated the reasonable simulation of depositions for $N_{ox}$ and $N_{rd}$ in MICS-Asia III.
**4.2 Contributions to the total inorganic N depositions and their potential effects**
Total inorganic N deposition (TIN), which includes the reduced and oxidized forms of
inorganic N deposition both from wet and dry processes, has been calculated for
estimating its ecosystem effects in this study as they were measured in most cases
before (Pan et al., 2012;Liu et al., 2013). Figure 10 and Figure 11 showed the pathway
of each type of N deposition to the TIN from spatial distribution view and 6 regions
statistical results, respectively. The ENM dry depositions of gaseous $HNO_3$ and $NH_3$
were the two major contributors to the TIN, both of which took part in 18% of TIN
over the whole country; while the wet deposition of $NO_3^-$ and $NH_4^+$ were another two
main components with the percentage of 23% and 28% (Table 4), respectively.
Consistent with that reported in the global assessment under HTAP (Vet et al., 2014)
and in nationwide monitoring network (NNDMN) estimation (Xu et al., 2015), the $N_{rd}$
in China dominated the TIN deposition with the averaged percentage reached at 52%
for the ensemble results, although slightly lower compared with 60% and 58% in the





two previous works. The overall contribution of wet and dry deposition to TIN was
almost half by half, which is consistent with that reported in NNDMN by Xu et al.
(Xu et al., 2015). Considering the total emission, the depositions in whole China took
about 67%, 65% and 66% in the 2010 emission of $NH_3$, $NO_x$ and total N
($NH_3$-N+$NO_x$-N), respectively. It is interesting to show that the relationship of the
gridded averaged $N_{rd}$ deposition as well as the $N_{ox}$ deposition with their relevant
emissions in six regions (shown in Figure 12 with the slope: 0.56, $r^2$:0.97 for $N_{rd}$ and
the slope: 0.47, $r^2$:0.88 for $N_{ox}$) were consistent with that reported by Xu et al. (Xu et
al., 2015) (slope: 0.51, $r^2$:0.89 for $N_{rd}$ and slope: 0.48, $r^2$:0.81 for $N_{ox}$). Even the
increasing trend of the regions from lowest in TP to highest in NC was the same as the
previously measurement study. This implicated the spatial distribution as well as the
relationships of deposition and emission were comparable with that measured in the
NNDMN. Pan et al. (2013) also compared the correlations of the observed depositions
vs emissions and attributed the inconsistent distribution between them in NCP to the
uncertainties of the emission. However, the patterns of depositions were also
influenced by the regional transport besides the emissions. In this study, significant
positive correlations of the simulated $N_{ox}$($N_{rd}$) depositions with the correspondingly
$NO_x$($NH_3$) emission reflected the control role of the relative emission to the spatial
distribution of the depositions. Although most regions were located below 1:1 line of
deposition to emission (Figure 12), few regions, such as TP and NE, were close to or
above 1:1 line implied the impacts of transport on deposition among the regions.

For regions, the area-averaged deposition of TIN was highest as 29.2 kg N•ha$^{-1}$ and
27 kg N•ha$^{-1}$ in NC and SE, followed by 15 kg N•ha$^{-1}$ and 10.1 kg N•ha$^{-1}$ in SW and
NE, respectively. The TIN in NW and TP were lowest as 3.1 and 2.7 kg N•ha$^{-1}$. In top
brand of two highest regions NC and SE, the deposition of TIN was similar but the
pathways to them were different. The $N_{rd}$ deposition (53%) and the dry deposition
(54%) contributed more than half portion of TIN in NC, while the $N_{ox}$ deposition
(55%) dominated the TIN in SE. Considering the lower ratio of $NO_x$/$NH_3$ emission in
SE (21.4/21.6, 0.99) than NC (30.4/24.4, 1.25), higher contribution of $N_{ox}$ to TIN in
SE indicated more oxidant ratio of the precursors than NC. For more oxidant N
species, i.e., $HNO_3$ and $NO_3^-$, both dry and wet depositions were higher in SE than
that shown in NC (5.8 vs. 4.9 for dry deposition of gaseous $HNO_3$ and 6.9 vs. 6.3 for
wet deposition of particulate $NO_3^-$). While for less oxidant N and the reduced N, all
type of depositions, such as dry deposition of gaseous $NO_x$, gaseous $NH_3$ as well as
the particulate $NH_4^+$, were less in SE than NC, except the wet deposition of particulate
$NH_4^+$, which would due to the much higher volume of rainfall in SE (Figure S5).
Overall, the oxidant N made the emitted $NO_x$ more easily to be scavenged in SE with
the ratio of $N_{ox}$-deposition/$NO_x$-emission reaching at 70%, while the reduced N is
more likely to be scavenged from its emission with the ratio as 64% in NC. The total
ratio of TIN/emission in NC and SE were 53% and 63%, respectively. Compared to
the Critical Load (Duan et al., 2001;Zhao et al., 2009;Liu et al., 2011), which is a
judgement of the deposited N effects to the ecosystem, the two regions were almost
reaching and even exceeding to the CL value (Table 4), indicating serious ecological
impacts of the N deposition in NC and SE and should be paid more attention to the


controlling of the N related species, especially the $N_{rd}$ in NC and $N_{ox}$ in SE.
In the less developed economic and social area of SW, Due to the high emission of
NH$_3$, 60% of the TIN was dominated by $N_{rd}$ deposition. The ratio of NO$_x$/NH$_3$
emission reached 0.49 as the more NH$_3$ emitted from agricultural activity than the
NO$_x$ from the fossil fuel consuming. The ratio of wet deposition/TIN was 55%, which
was lower than the HTAP comparison during 2000 (60-70%) (Vet et al., 2014) but
higher than the results of NNDMN (45%)(Xu et al., 2015). Although the undeveloped
society, the TIN deposition was almost reaching at the CL value according to Zhao et
al. (Zhao et al., 2009). Besides the high emission of NH$_3$, high ratio of
$N_{ox}$-deposition/NO$_x$-emission reaching at 94% reflecting the import of $N_{ox}$ from high
emitted area, such as SE and NC, should be attracted our attention in this region.
Although the N deposition in TP was not reaching at CL value, which was the lowest
in all regions of China with the value of 2.7 kg N•ha$^{-1}$, the N ecological impacts
cannot be neglected since the sensitive ecosystem (Shen et al., 2019) as well as the
important climatically influence to whole China. Considering the high ratio of
TIN/emission, which were larger than 1 with 3 for TIN, 2.71 for $N_{rd}$ and 4 for $N_{ox}$, the
import from outside was responsible to the N deposition in TP.
**5 Conclusion**
Reactive N depositions over China simulated in the frame work of MICS-Asia III
have been compared within each participant models. Wet depositions were also
validated by the multi-source of observations, i.e., recorded data from EANET, CAS,
NNDMN and EPA of Guangdong and Liaoning province. Most models show the
consistent spatial and temporal variation of both $N_{ox}$ and $N_{rd}$ wet depositions in China
with the NME around at 50%, which is lower than the value of 70% based on EANET
observation over Asia. Coefficient of Variation (CV) was applied for model
comparison of dry deposition as well as the related precursor's concentration in the air
mass. Consistence of both spatial and temporal variation of CV in deposition and the
concentration in air mass indicated that performance of the precursors' simulation was
highly correlated with their depositions.
Large deposition of ensemble simulation of $N_{rd}$ deposition in eastern China was
corresponding to low level of VCD from satellite measurement, while the case of $N_{ox}$
was just on the contrary. The total emission of NO$_x$ and NH$_3$ was similar at 8 kg
N•ha$^{-1}$ in China. This indicated the allocation of both NO$_x$ and NH$_3$ between the
deposition to the surface ground and staying in the atmosphere were conserved to
their emission into the air, which also implicated the reasonable simulation of
depositions for $N_{ox}$ and $N_{rd}$ in MICS-Asia III.
Wet deposition of nitrate and ammonium as well as the dry deposition of Gaseous
NH$_3$ and HNO$_3$ were the important pathway to TIN deposition with the percentage as
18%, 18%, 23% and 28% for ensemble results, respectively. The gridded averaged $N_{rd}$
in China dominated the TIN deposition with the averaged percentage reached at 52%,
which was slightly lower than the reported 60% and 58% in HTAP and NNDMN
measurements. The contribution of wet and dry deposition to TIN was almost half by
half and consistent with that reported in NNDMN. Even the ratio of TIN/emission
was also similar with the NNDMN, indicating that the spatial distribution as well as





the relationships of deposition and emission were comparable with that measured in the NNDMN.

For different regions of China, the simulated $N_{ox}$ wet deposition was overestimated in NE but underestimated in SE and SW, while large uncertainties were shown in NC. Two models simulated 30-50% higher of $N_{ox}$ wet deposition, and four models were 20~40% lower compared with observations in NC. The large divergences do not only exist between different models but also in the same CMAQ model, i.e., M1-M6. For the simulation of $N_{rd}$ wet deposition, all the models were underpredicted in all regions, with the largest underestimation in NC and SE. Different from $N_{ox}$, almost similar behavior prediction of the less oxidative species such as the $N_{rd}$ wet deposition has been found in CMAQ models, indicating the chemical reaction process is the one of the most importance factors affecting the model uncertainties in MICS-Asia III. Compared to CL value, the reactive N deposition in NC, SE and SW reached or exceeded the reported CL value and indicated serious ecological impacts. The control of $N_{rd}$ in NC and SW and $N_{ox}$ in SE would be effective to mitigate the TIN deposition in these regions. For the lowest reactive N deposition in TP, however, the N ecological impacts cannot be neglected since the sensitive ecosystem as well as the important climatically influence to whole China, especially considering the high ratio of TIN/emission, which was mainly caused by the import from outside. The joint prevention and control of air pollution in China should be carefully considered and implemented in the future.

**Acknowledgment**

We appreciate the Guangdong and Liaoning EPA for providing the observation data of Guangdong and Liaoning province. This work is supported by the National Natural Science Foundation of China (Grant No 41571130024, 41575123, 91744206, 41330422) and the National Key Research and Development Plan (20017YFC0210100).





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



**Tables**

Table 1. Mechanism and parameterization of deposition part of MICS-Asia III

| No | M1 | M2 | M4 | M5 | M6 | M11 | M12 | M13 | M14 |
|---|---|---|---|---|---|---|---|---|---|
| Model-(version)[a] | CMAQ v5.0.2 | CMAQ v5.0.2 | CMAQ v4.7.1 | CMAQ v4.7.1 | CMAQ v4.7.1 | NAQPMS | NHM-Chem | GEOS-Chem | CMAQ v4.7.1 |
| Advection-H[b] | Yamo | Yamo | PPM | PPM | Yamo | WA | WA | TPCORE | PPM |
| Advection-V[b] | PPM | PPM | PPM | PPM | Yamo | WA | WA | TPCORE | PPM |
| Diffusion-H[b] | multiscale | multiscale | multiscale | multiscale | multiscale | BD | multiscale | HB | multiscale |
| Diffusion-V[b] | ACM2 | ACM2 | ACM2 | ACM2 | ACM2 | K-theory | MYJ | HB | ACM2 |
| Gas-Chemistry[c] | SAPRC-99 | SAPRC-99 | SAPRC-99 | SAPRC-99 | SAPRC-99 | CBMZ | SAPRC-99 | Bey | SAPRC-99 |
| Aerosol-chemistry[d] | AERO6 ISORROPIA (v2.1) | AERO6 ISORROPIA (v2.1) | AERO5 ISORROPIA (v1.7) | AERO5 ISORROPIA (v1.7) | AERO5 ISORROPIA (v1.7) | ISORROPIA (v1.7) | ISORROPIA (v2.1) | ISORROPIA (2.1) | ISORROPIA (v1.7) |
| Cloud & Aqueous[e] | ACM-AE6 | ACM-AE6 | ACM-AE5 | ACM_AE5 | ACM_AE5 | Ge | WC | Jacob | ACM |
| Dry dep[f] | M3DRY | M3DRY | M3DRY | M3DRY | M3DRY | Wesely | Kajino | Wesely & Wang | M3DRY |
| Wet dep[g] | Foley | Foley | Foley | Foley | ACM | Ge | Kajino | Liu | Foley |
| Met | WRF | WRF | WRF | WRF | WRF | WRF | WRF | GEOS-5 | RAMS |
| Emission[h] | standard | standard | standard | standard | standard | standard | standard | standard | standard |

[a]: References for the advection scheme are as follows: Yamo:Yamartino, 1993; PPM: Piecewise Parabolic Method (Colella and Woodward, 1984); WA: Walcek and Aleksic, 1998; TPCORE: Wang et al., 2004.

[b]: References for diffusion scheme are as follows: ACM2: Asymmetric Convective Model version 2 (Pleim, 2007a,b); BD: Byun and Dennis, 1995; HB: Holtslag and Boville, 1993; multiscale: Byun and Schere, 2006; MYJ: Janjic, 1994.



[c]: References for the gas phase chemistry are as follows: Bey: Bey et al., 2001; CBMZ: Zaveri and Peters, 1999; SAPRC-99: Carter, 2000.
[d]: References for the aerosol chemistry are as follows: ISORROPIA version 2.1: Fountoukis and Nenes, 2007; version 1.7: Nenes et al., 1998.
[e]: References for the Cloud & Aqueous are as follows: Ge: Ge et al., 2014; WC: Walcek,1986 and Carlton, 2007; Jacob: Jacob, 2000;
[f]: References for the dry deposition scheme are as follows: M3DRY: Pleim et al., 2001; Kajino: Kajino et al., 2018; Wang: Wang et al., 2004;
Wesely: Wesely, 1989.
[g]: References for the wet deposition scheme are as follows: Foley: Foley et al, 2010; Ge: Ge et al., 2014; Kajino: Kajino et al., 2018; Liu: Liu et
al., 2001.
[h]. "standard" indicates the basic emission inventories in Phase III.







Table 2. Statistical parameters of oxidized N deposition for urban, rural and whole China.

| Urban (N=49) | OBS | M1 | M2 | M4 | M5 | M6 | M11 | M12 | M13 | M14 |
|---|---|---|---|---|---|---|---|---|---|---|
| | | | | | Oxidized N deposition (kg N ha$^{-1}$) | | | | | |
| R | | 0.24 | 0.25 | 0.28 | 0.27 | 0.26 | 0.25 | 0.40 | 0.37 | 0.22 |
| NMB% | | 16.8% | -44.5% | -19.3% | 38.5% | -15.8% | 48.2% | -5.3% | -7.2% | -52.7% |
| NME% | | 56.4% | 60.4% | 51.3% | 64.0% | 51.0% | 67.1% | 46.9% | 44.2% | 59.1% |
| FAC2% | | 63.3% | 32.7% | 51.0% | 57.1% | 57.1% | 59.2% | 57.1% | 61.2% | 40.8% |
| Mean | 7.1 | 8.3 | 3.9 | 5.7 | 9.9 | 6.0 | 10.5 | 6.7 | 6.6 | 3.4 |
| | | | | | Oxidized N concentration in rainfall (mg N/L) | | | | | |
| R | | 0.60 | 0.57 | 0.60 | 0.62 | 0.59 | 0.49 | 0.61 | 0.52 | 0.50 |
| NMB% | | 26.8% | -37.9% | -11.3% | 49.3% | -6.7% | 75.9% | 2.7% | 19.3% | -31.3% |
| NME% | | 57.6% | 51.4% | 46.0% | 69.0% | 47.5% | 94.8% | 47.4% | 65.5% | 60.0% |
| FAC2% | | 59.2% | 42.9% | 59.2% | 51.0% | 59.2% | 51.0% | 61.2% | 49.0% | 34.7% |
| Mean | 0.9 | 1.1 | 0.5 | 0.8 | 1.3 | 0.8 | 1.5 | 0.9 | 1.0 | 0.6 |

| Rural (N=34) | OBS | M1 | M2 | M4 | M5 | M6 | M11 | M12 | M13 | M14 |
|---|---|---|---|---|---|---|---|---|---|---|
| | | | | | Oxidized N deposition (kg N ha$^{-1}$) | | | | | |
| R | | 0.09 | 0.05 | 0.09 | 0.14 | 0.09 | 0.28 | 0.26 | 0.23 | 0.30 |
| NMB% | | 55.4% | -27.8% | 5.1% | 86.9% | 9.9% | 102.5% | 11.5% | 13.6% | -37.6% |
| NME% | | 83.7% | 57.8% | 59.9% | 103.3% | 60.5% | 110.1% | 54.4% | 56.0% | 50.3% |
| FAC2% | | 55.9% | 41.2% | 50.0% | 35.3% | 47.1% | 38.2% | 55.9% | 55.9% | 50.0% |
| Mean | 5.4 | 8.5 | 3.9 | 5.7 | 10.2 | 6.0 | 11.0 | 6.1 | 6.2 | 3.4 |
| | | | | | Oxidized N concentration in rainfall (mg N/L) | | | | | |
| R | | 0.43 | 0.41 | 0.44 | 0.46 | 0.44 | 0.48 | 0.47 | 0.35 | 0.43 |
| NMB% | | 20.5% | -43.0% | -17.1% | 45.2% | -13.4% | 63.2% | -9.4% | -0.2% | -43.2% |
| NME% | | 65.4% | 55.6% | 54.2% | 76.3% | 53.8% | 89.2% | 53.8% | 62.7% | 53.3% |
| FAC2% | | 44.1% | 38.2% | 41.2% | 41.2% | 44.1% | 41.2% | 47.1% | 32.4% | 41.2% |







| | | | | | | | | | | |
|---|---|---|---|---|---|---|---|---|---|---|
| Mean | 0.9 | 1.0 | 0.5 | 0.7 | 1.2 | 0.7 | 1.4 | 0.8 | 0.9 | 0.5 |
| **All sites (N=83)** — Oxidized N deposition (kg N ha⁻¹) | | | | | | | | | | |
| R | | 0.2 | 0.17 | 0.21 | 0.21 | 0.19 | 0.24 | 0.37 | 0.33 | 0.23 |
| NMB% | | 30.3% | -38.7% | -10.7% | 55.4% | -6.8% | 67.2% | 0.6% | 0.1% | -47.4% |
| NME% | | 66.0% | 59.5% | 54.3% | 77.8% | 54.3% | 82.2% | 49.5% | 48.3% | 56.0% |
| FAC2% | | 60.2% | 36.1% | 50.6% | 48.2% | 53.0% | 50.6% | 56.6% | 59.0% | 44.6% |
| Mean | 6.4 | 8.4 | 3.9 | 5.7 | 10.0 | 6.0 | 10.7 | 6.5 | 6.4 | 3.4 |
| Oxidized N concentration in rainfall (mg N/L) | | | | | | | | | | |
| R | | 0.53 | 0.51 | 0.54 | 0.56 | 0.53 | 0.48 | 0.56 | 0.46 | 0.46 |
| NMB% | | 24.2% | -40.0% | -13.7% | 47.6% | -9.5% | 70.7% | -2.3% | 11.3% | -36.2% |
| NME% | | 60.8% | 53.1% | 49.4% | 72.0% | 50.1% | 92.5% | 50.0% | 64.4% | 57.3% |
| FAC2% | | 53.0% | 41.0% | 51.8% | 47.0% | 53.0% | 47.0% | 55.4% | 42.2% | 37.3% |
| Mean | 0.9 | 1.1 | 0.5 | 0.7 | 1.3 | 0.8 | 1.5 | 0.8 | 1.0 | 0.5 |





Table 3. Same as Table 2 but for reduced N deposition.

| Urban (N=49) | OBS | M1 | M2 | M4 | M5 | M6 | M11 | M12 | M13 | M14 |
|---|---|---|---|---|---|---|---|---|---|---|
| | | | | Reduced N deposition (kg N ha$^{-1}$) | | | | | | |
| R | | 0.30 | 0.31 | 0.33 | 0.34 | 0.32 | 0.41 | 0.33 | 0.49 | 0.05 |
| NMB% | | -38.2% | -43.0% | -45.6% | -43.9% | -37.3% | -73.5% | -38.8% | -38.8% | -60.2% |
| NME% | | 50.7% | 52.2% | 52.9% | 51.4% | 49.8% | 73.5% | 50.0% | 46.3% | 64.1% |
| FAC2% | | 53.1% | 44.9% | 42.9% | 46.9% | 51.0% | 16.3% | 51.0% | 55.1% | 34.7% |
| Mean | 10.9 | 6.7 | 6.2 | 5.9 | 6.1 | 6.8 | 2.9 | 6.7 | 6.7 | 4.3 |
| | | | | Reduced N concentration in rainfall (mg N/L) | | | | | | |
| R | | 0.83 | 0.83 | 0.84 | 0.84 | 0.83 | 0.77 | 0.86 | 0.75 | 0.56 |
| NMB% | | -38.0% | -42.6% | -42.2% | -40.6% | -36.1% | -73.8% | -41.1% | -22.1% | -48.8% |
| NME% | | 44.5% | 47.5% | 47.4% | 46.0% | 43.7% | 73.8% | 43.7% | 46.0% | 62.1% |
| FAC2% | | 57.1% | 51.0% | 42.9% | 51.0% | 55.1% | 10.2% | 57.1% | 44.9% | 24.5% |
| Mean | 1.5 | 0.9 | 0.9 | 0.9 | 0.9 | 1.0 | 0.4 | 0.9 | 1.2 | 0.8 |
| **Rural (N=34)** | | | | | | | | | | |
| | | | | Reduced N deposition (kg N ha$^{-1}$) | | | | | | |
| R | | 0.29 | 0.29 | 0.28 | 0.30 | 0.32 | 0.27 | 0.28 | 0.52 | 0.44 |
| NMB% | | -14.4% | -22.0% | -21.1% | -18.3% | -13.6% | -62.5% | -23.2% | -19.0% | -46.2% |
| NME% | | 48.0% | 47.7% | 48.1% | 46.5% | 47.2% | 68.5% | 45.8% | 40.8% | 49.4% |
| FAC2% | | 70.6% | 55.9% | 58.8% | 61.8% | 67.6% | 23.5% | 61.8% | 73.5% | 52.9% |
| Mean | 9.0 | 7.7 | 7.0 | 7.1 | 7.3 | 7.7 | 3.4 | 6.9 | 7.3 | 4.8 |
| | | | | Reduced N concentration in rainfall (mg N/L) | | | | | | |
| R | | 0.79 | 0.79 | 0.81 | 0.82 | 0.80 | 0.74 | 0.82 | 0.69 | 0.55 |
| NMB% | | -27.5% | -34.2% | -31.9% | -29.7% | -26.4% | -69.0% | -33.8% | -20.2% | -47.4% |
| NME% | | 37.7% | 40.6% | 39.0% | 36.6% | 37.1% | 69.6% | 37.9% | 40.5% | 56.9% |
| FAC2% | | 52.9% | 52.9% | 52.9% | 52.9% | 55.9% | 17.6% | 52.9% | 64.7% | 44.1% |





| | Mean | | | | | | | | | |
|---|---|---|---|---|---|---|---|---|---|---|
| | 1.3 | 1.0 | 0.9 | 0.9 | 0.9 | 1.0 | 0.4 | 0.9 | 1.1 | 0.7 |
| **All sites (N=83)** | | | | | | | | | | |
| | | | | | Reduced N deposition (kg N ha⁻¹) | | | | | |
| R | | 0.26 | 0.27 | 0.26 | 0.27 | 0.28 | 0.30 | 0.29 | 0.48 | 0.20 |
| NMB% | | -29.6% | -35.3% | -36.7% | -34.5% | -28.6% | -69.5% | -33.1% | -31.6% | -55.1% |
| NME% | | 49.7% | 50.6% | 51.2% | 49.6% | 48.9% | 71.7% | 48.5% | 44.3% | 58.7% |
| FAC2% | | 60.2% | 49.4% | 49.4% | 53.0% | 57.8% | 19.3% | 55.4% | 62.7% | 42.2% |
| Mean | 10.1 | 7.1 | 6.5 | 6.4 | 6.6 | 7.2 | 3.1 | 6.8 | 6.9 | 4.5 |
| | | | | | Reduced N concentration in rainfall (mg N/L) | | | | | |
| R | | 0.81 | 0.81 | 0.82 | 0.83 | 0.81 | 0.75 | 0.84 | 0.73 | 0.56 |
| NMB% | | -34.0% | -39.4% | -38.2% | -36.4% | -32.3% | -71.9% | -38.3% | -21.3% | -48.3% |
| NME% | | 41.9% | 44.9% | 44.2% | 42.3% | 41.2% | 72.2% | 41.5% | 43.9% | 60.1% |
| FAC2% | | 55.4% | 51.8% | 47.0% | 51.8% | 55.4% | 13.3% | 55.4% | 53.0% | 32.5% |
| Mean | 1.4 | 0.9 | 0.9 | 0.9 | 0.9 | 1.0 | 0.4 | 0.9 | 1.1 | 0.7 |





Table 4. Types of depositions and its relevant contributions to TIN as well as the
emissions of reduced and oxidized N in different regions (Unit: kg N/ha/yr).

| Regions | | NC | NE | NW | SE | SW | TP | China |
|---|---|---|---|---|---|---|---|---|
| Types of deposition | gHNO3d | 4.9 | 1.8 | 0.8 | 5.8 | 2.4 | 0.2 | 2.1 |
| | gNH3d | 6.7 | 1.8 | 0.5 | 3.7 | 3.0 | 0.5 | 2.0 |
| | gNOxd | 1.2 | 0.3 | 0.1 | 1.0 | 0.3 | 0.0 | 0.3 |
| | pNH4d | 1.9 | 0.5 | 0.2 | 1.5 | 0.8 | 0.1 | 0.6 |
| | pNO3d | 1.3 | 0.4 | 0.1 | 1.2 | 0.4 | 0.0 | 0.4 |
| | pNH4w | 7.0 | 2.6 | 0.8 | 7.0 | 5.2 | 1.3 | 3.2 |
| | pNO3w | 6.3 | 2.7 | 0.7 | 6.9 | 3.0 | 0.6 | 2.6 |
| | $N_{rd}$ | 15.6 | 4.9 | 1.6 | 12.2 | 9.0 | 1.9 | 5.9 |
| | $N_{ox}$ | 13.6 | 5.2 | 1.6 | 14.9 | 6.0 | 0.8 | 5.4 |
| | Wet TIN | 13.3 | 5.3 | 1.5 | 13.9 | 8.2 | 1.9 | 5.8 |
| | Dry TIN | 16.0 | 4.8 | 1.7 | 13.2 | 6.9 | 0.8 | 5.5 |
| | TIN | 29.2 | 10.1 | 3.1 | 27.0 | 15.0 | 2.7 | 11.3 |
| $N_{rd}$/TIN % | This study | 53 | 49 | 52 | 45 | 60 | 70 | 52 |
| | NNDMN | | | | | | | 58 |
| | HTAP | | | | | | | >60 |
| Wet/TIN % | This study | 46 | 52 | 48 | 51 | 55 | 70 | 51 |
| | NNDMN | 43 | 46 | 39 | 58 | 45 | 50 | 48 |
| | HTAP | 40~50 | 40~60 | 30~60 | ~60 | 60~70 | 60~70 | |
| Emission | $N_{rd}$ | 24.4 | 4.9 | 2.9 | 21.6 | 13.1 | 0.7 | 8.7 |
| | $N_{ox}$ | 30.4 | 5.6 | 3.1 | 21.4 | 6.4 | 0.2 | 8.3 |
| | TIN | 54.8 | 10.5 | 5.9 | 43.0 | 19.5 | 0.9 | 17.1 |
| Dep/Emi % | $N_{rd}$ | 64 | 100 | 55 | 56 | 69 | 271 | 67 |
| | $N_{ox}$ | 45 | 93 | 52 | 70 | 94 | 400 | 65 |
| | TIN | 53 | 96 | 53 | 63 | 77 | 300 | 66 |
| Critical load | SSMB1 | 10~30 | 5~20 | <5 | 10~20 | >20 | 10~15 | |
| | Empirical* | >200 | <15 | <15 | 50~200 | 50~200 | 20~50 | |
| | SSMB2 | >50 | 14-50 | <14 | 20-50 | 10~30 | ~14 | |

1, Duan et al.,2, Zhao et al.,* Liu et al.,,



**Figures and captions**

**Figure 1:** Locations of the measurement sites and the distribution of the ID.

**Figure 2:** Percentile Box plot of oxidized N wet deposition simulated in each model and compared with the observation as well as the rainfalls, with 99% and 1% represented for the top and low points, 90% and 10% represented for the top and low horizontal line, 75% and 25% represented for the upper and lower edge of the box and asterisk and long horizontal line in the middle of the box represented for the medium and mean value, respectively.

**Figure 3:** Monthly variation of simulated wet deposition of oxidized N compared with the observations in urban sites (a) and rural sites (b) of NC; urban sites (c) and rural sites (d) of NE; as well as of reduced N in urban sites (e) and rural sites (f) of NC; urban sites (g) and rural sites (h) of NE.

**Figure 4:** Same as Figure 2 but for reduced N wet depositions.

**Figure 5:** Distributions of the wet depositions of $N_{ox}$ simulated by M1~M14 (a)~(i), ENM of the multi-models (j) MICS-Asia III, observation from multi source measurements (k) and the comparison between ENM and observations (l) (kgN•ha$^{-1}$).

**Figure 6:** Same as Figure 5 but for $N_{rd}$.

**Figure 7:** Spatial distribution of CV of (a) $N_{ox}$ dry deposition, (b) $N_{rd}$ dry deposition, (c) $N_{ox}$ wet deposition and (d) $N_{rd}$ wet deposition in MICS-Asia III on the annual and seasonal basis.

**Figure 8:** Distribution of CV of $NO_x$ (a), $NO_3^-$ (b), $NH_3$ (c) and $NH_4^+$ (d) in the air mass for seasonal and annual.

**Figure 9:** ENM results for dry deposition (a) and wet deposition (b) of $N_{ox}$, VCD of $NO_2$ from SCIAMACHY (c) and $NO_x$ emission from MICS-Asia (d); ENM results for dry deposition (e) and wet deposition (f) of $N_{rd}$, VCD of $NH_3$ from IASI (g) and $NH_3$ emission from MICS-Asia (h).

**Figure 10:** ENM results of each process of N deposition flux (a) and the fraction in TIN (b) in MICS-Asia III. The icons w_N, w_A, d_NO2, d_NH3, d_HNO3, d_ammonium and d_nitrate represented wet deposition of nitrate, wet deposition of ammonium, dry deposition of $NH_3$, dry deposition of $HNO_3$, dry deposition of ammonium and dry deposition of nitrate, respectively.

**Figure 11:** Pathway of N species to TIN deposition in different regions from ENM results (a), and TIN depositions by wet or dry deposited manner (b) in percentile Box plot; with 90% and 10% represented for the top and low horizontal line, 75% and 25% represented for the upper and lower edge of the box and asterisk in the middle of the





box represented for the medium value, respectively.
**Figure 12:** Relationship of $N_{rd}$ deposition vs. $NH_3$ emission (a) and relationship of
$N_{ox}$ deposition vs. $NO_x$ emission (b) in each region of China.


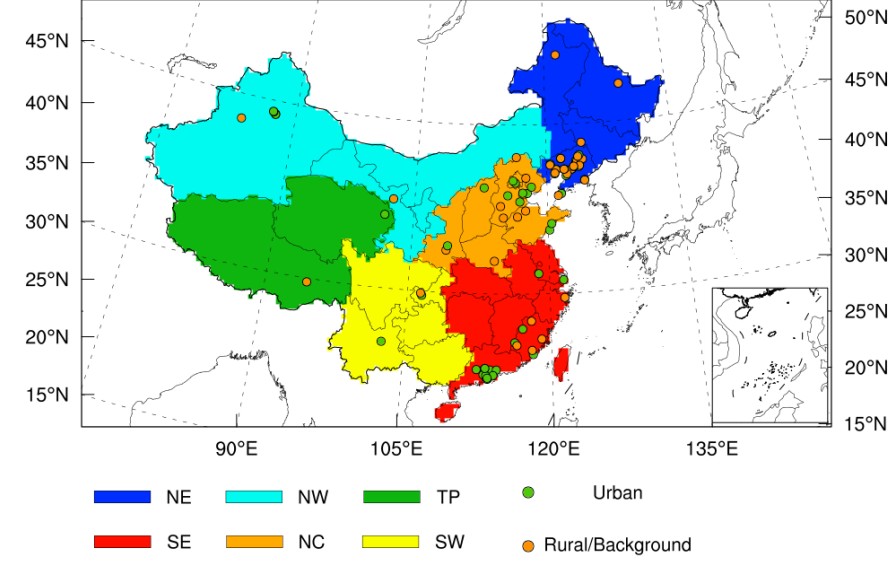


Figure 1: Locations of the measurement sites and the distribution of the ID




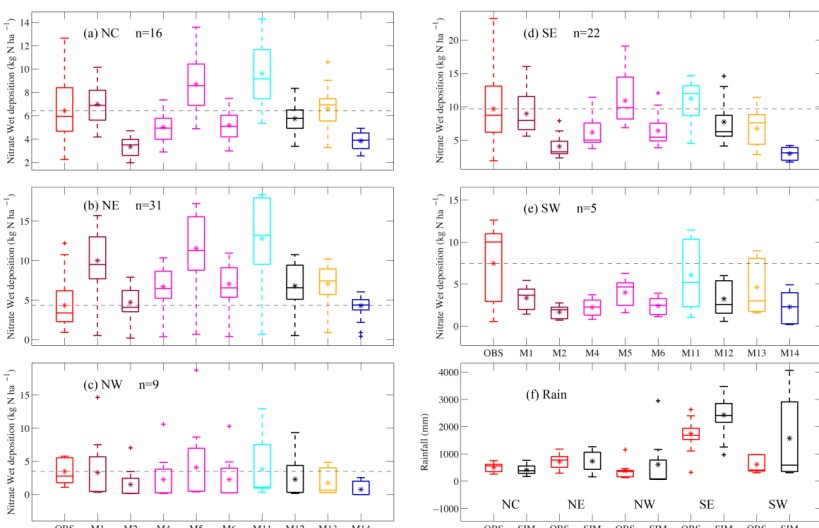


Figure 2: Percentile Box plot of oxidized N wet deposition simulated in each model
and compared with the observation as well as the rainfalls, with 99% and 1%
represented for the top and low points, 90% and 10% represented for the top and low
horizontal line, 75% and 25% represented for the upper and lower edge of the box and
asterisk and long horizontal line in the middle of the box represented for the medium
and mean value, respectively



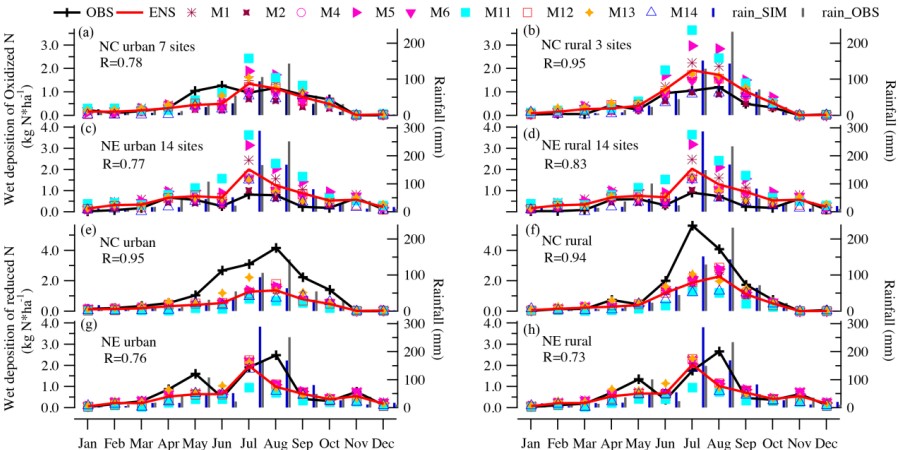


Figure 3: Monthly variation of simulated wet deposition of oxidized N compared with
the observations in urban sites (a) and rural sites (b) of NC; urban sites (c) and rural
sites (d) of NE; as well as of reduced N in urban sites (e) and rural sites (f) of NC;
urban sites (g) and rural sites (h) of NE






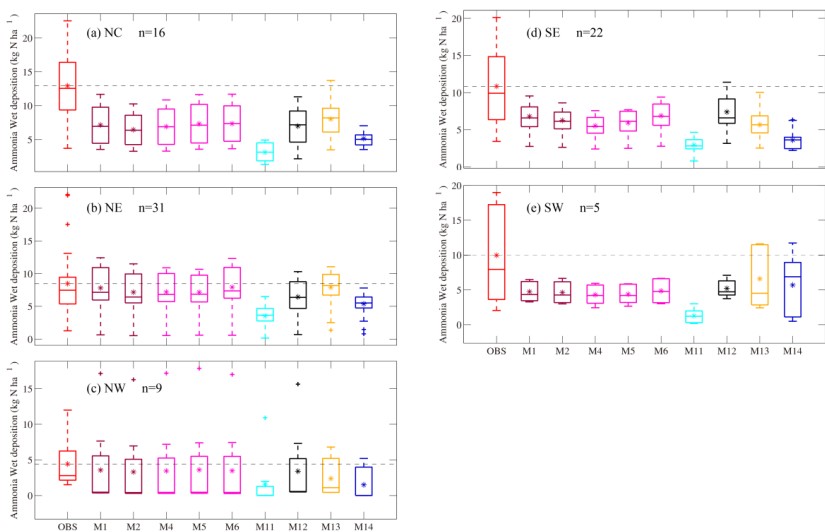


Figure 4: Same as Figure 2 but for reduced N wet depositions



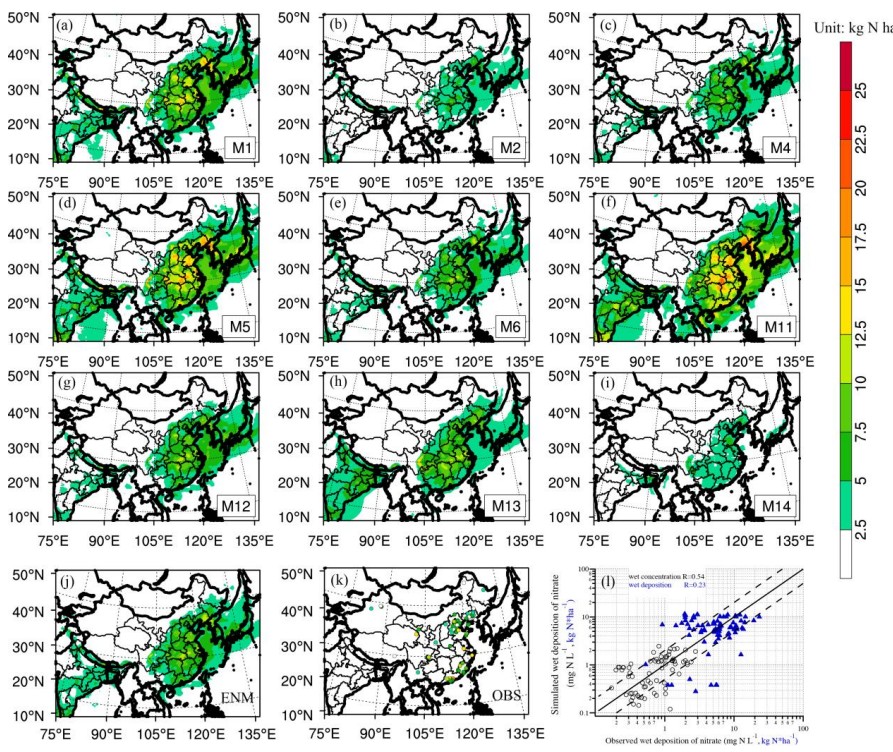

Figure 5: Distributions of the wet depositions of $N_{ox}$ simulated by M1~M14 (a)~(i), ENM of the multi-models (j) MICS-Asia III, observation from multi source measurements (k) and the comparison between ENM and observations (l) (kgN•ha$^{-1}$)




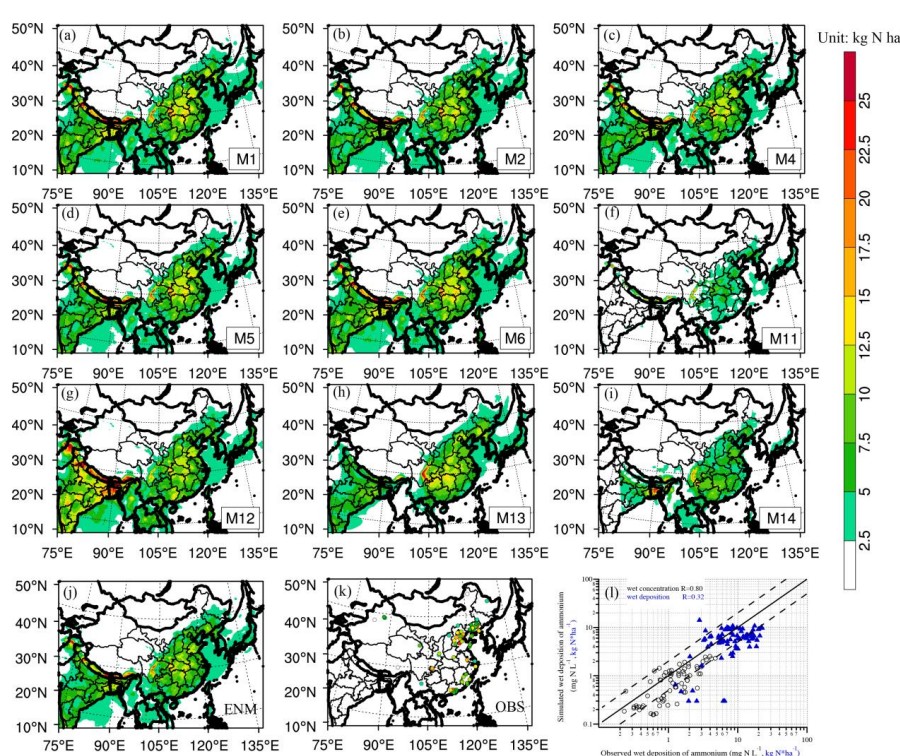


Figure 6: Same as Figure 5 but for $N_{rd}$




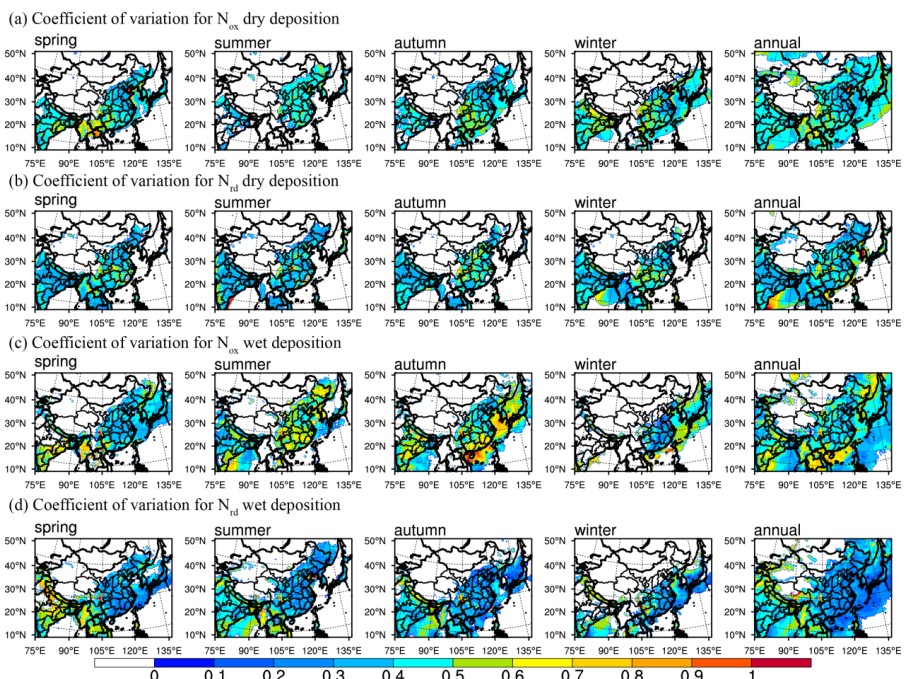

Figure 7: Spatial distribution of CV of (a) $N_{ox}$ dry deposition, (b) $N_{rd}$ dry deposition,
(c) $N_{ox}$ wet deposition and (d) $N_{rd}$ wet deposition in MICS-Asia III on the annual and
seasonal basis






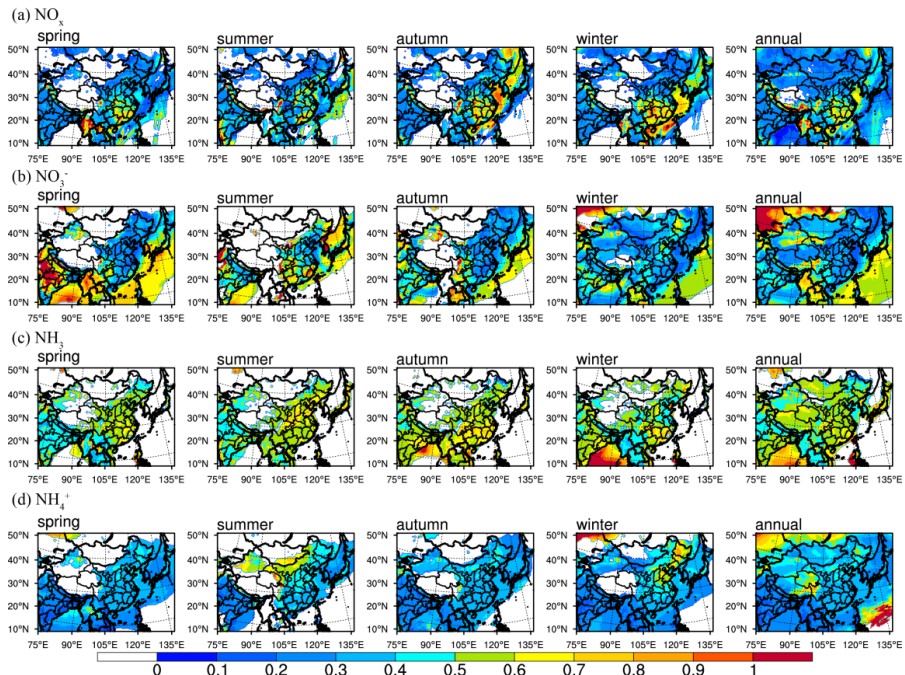


Figure 8: Distribution of CV of $NO_x$ (a), $NO_3^-$ (b), $NH_3$ (c) and $NH_4^+$ (d) in the air
mass for seasonal and annual


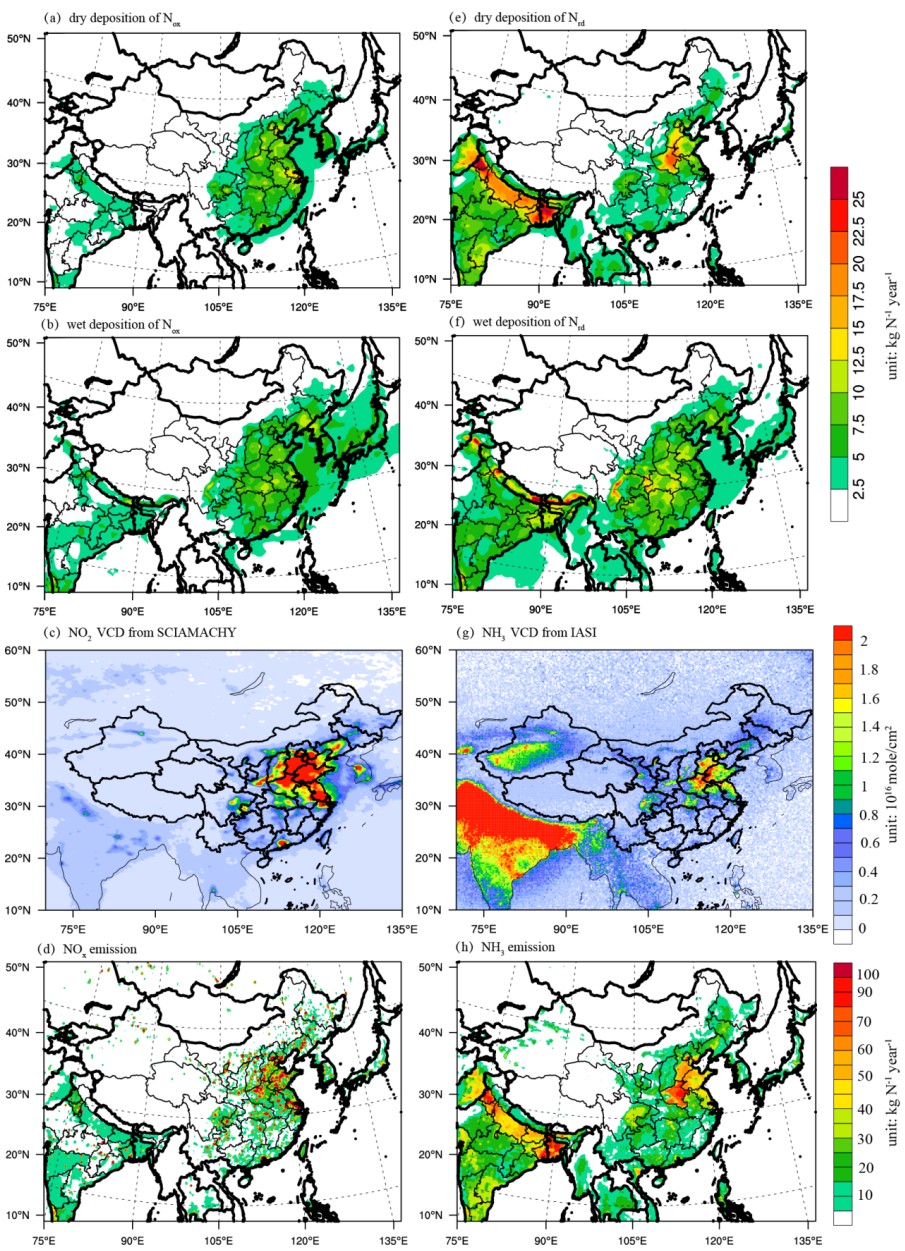

Figure 9: ENM results for dry deposition (a) and wet deposition (b) of $N_{ox}$, VCD of $NO_2$ from SCIAMACHY (c) and $NO_x$ emission from MICS-Asia (d); ENM results for dry deposition (e) and wet deposition (f) of $N_{rd}$, VCD of $NH_3$ from IASI (g) and $NH_3$ emission from MICS-Asia (h)

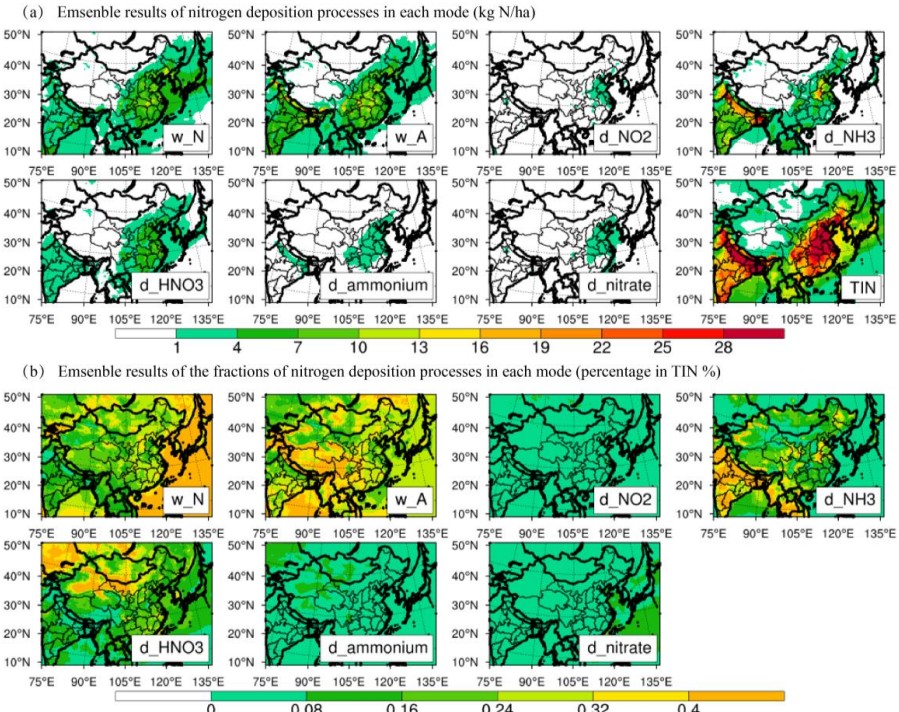

1083

Figure 10: ENM results of each process of N deposition flux (a) and the fraction in TIN (b) in MICS-Asia III. The icons w_N, w_A, d_NO2, d_NH3, d_HNO3, d_ammonium and d_nitrate represented wet deposition of nitrate, wet deposition of ammonium, dry deposition of NH₃, dry deposition of HNO₃, dry deposition of ammonium and dry deposition of nitrate, respectively.



1090

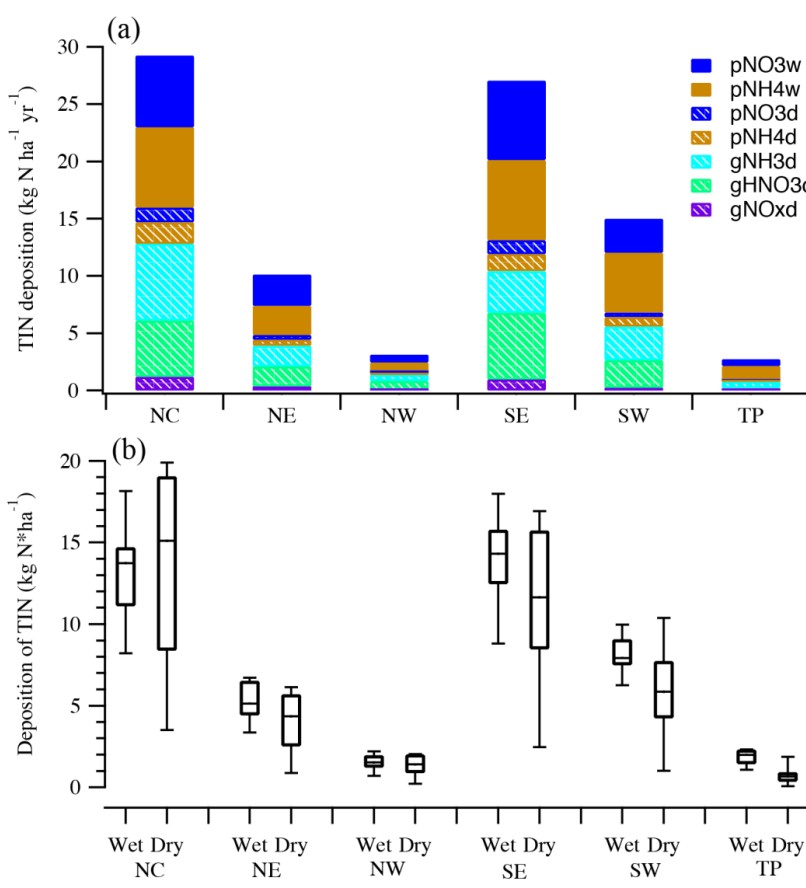

1091

Figure 11: Pathway of N species to TIN deposition in different regions from ENM results (a), and TIN depositions by wet or dry deposited manner (b) in percentile Box plot; with 90% and 10% represented for the top and low horizontal line, 75% and 25% represented for the upper and lower edge of the box and asterisk in the middle of the box represented for the medium value, respectively




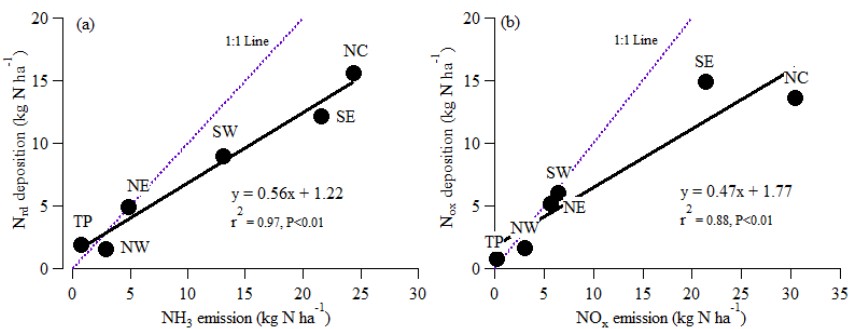


Figure 12: Relationship of $N_{rd}$ deposition vs. $NH_3$ emission (a) and relationship of $N_{ox}$
deposition vs. $NO_x$ emission (b) in each region of China