# Peer review of "MICS-Asia III: Multi-model comparison of reactive Nitrogen deposition"

_Atmospheric Chemistry and Physics, 2019_

## Referee Comment (RC1) · Anonymous Referee #2 · 28 Apr 2020

This manuscript examines the performance of nine models within MICS-Asia III in capturing wet and dry deposition with observation from a plenty of sites. There are a number of grammar issues and typos. I would enough the authors to check carefully and improve the quality of writing. Besides, the scientific quality would be largely improved if the authors could provide more results to explain the differences between models and observations. Specific comments are listed below:

Line 52: Typo: WRF-CAMQ should be WRF-CMAQ

Line 105: Typo: shown should be showed

Line 123: uncompleted should be incomplete

Line 179: driving should be driven

[Figure]

Line 292: may not due to should be "may not be due to"

Lines 292-294: What would be the reason for the differences?

In many places, wrong tense is used. For example, in line 295: showed should be shows. Please also change the tense in other places.

Line 312-314: The differences might result from multiple reasons, including emissions, chemical conversions, deposition processes in the models, etc. There is no evidence or analysis showing that is caused by the coarse grid.

Lines 335-343: This could be one of the reasons with the assumption that models can accurately capture these processes. Is it possible to compare rainfall events, which does not require high resolution of deposition data

Lines 575-576: correlations between observed depositions between emissions

Line 662: importance should be important

---

## Referee Comment (RC2) · Anonymous Referee #1 · 4 May 2020

This manuscript has presented the analyses of atmospheric nitrogen deposition to China as simulated by an ensemble of chemical transport models participating the MICS-Asia III model intercomparison. Available surface measurements of wet deposition fluxes are integrated to assess the model performances. This represents an important step towards a better understanding the model range and uncertainties in simulating nitrogen deposition. Different from previous studies on multi-model nitrogen deposition simulation, most models analyzed in this study used the same emissions and driving meteorology, allowing a closer attribution of the factors driving the model uncertainties. The results show that most models calculated consistent spatial and temporal variations of nitrogen deposition for both oxidized and reduced nitrogen, yet considerable differences exist among models.

[Figure]

I think the study is an important contribution to the MICS-Asia III special issue. The analyses are mostly fine, and it would be much scientifically stronger if having a deeper investigation on the drivers of model differences. The following comments also need to be addressed. In addition to my specific comments as elaborated below, improvements on the language are necessary and need caution.

**Specific comments:**
1. Page 5, Line 199-202: How about natural sources of nitrogen, e.g., nitrogen oxides from soil and lightning? Are they included in any of these models?

2. Page 6, Line 220-223:
Did any of the models also simulate dry deposition of other nitrogen species, e.g., PAN, isoprene nitrates? How important are these nitrogen species contributing to dry deposition, and the uncertainty induced by excluding them in the analysis? Please clarify.

3. Page 7, Line 261:
It is not clear how the data are normalized as monthly wet deposition fluxes. Do you mean that the raw measurements are at different temporal resolutions (daily, weekly, etc.), and then are interpolated to monthly values? This shall be explained in the text.

4. Page 7, Line 279-281:
This sentence is not clear. Do you mean the correlation coefficients are lower than the value when only comparing with EANET data? Please clarify.

5. Page 7, Line 285-286:
Need to add a sentence defining FAC2.

6. Page 10, Line 417-423:

The sentence here needs rewritten or removed. It is a long sentence, and the information is repetitive in the paragraph.

7. Page 12, Line 478-480:
Is Figure 8 for the surface layer or the atmospheric column? As dry deposition only applies to species at the surface layer, while wet deposition can extend to the whole tropospheric column, an explanation is needed here to justify why you use it for both dry and wet depositions.

8. Page 12, Line 500-510:
How about dry deposition velocities? Did all the models calculate the dry deposition fluxes as the products of surface concentration and dry deposition velocity? It is missing something that the discussion of dry deposition only examined concentrations and not include dry deposition velocities.

9. Page 13, Line 538-548:
The discussion of different allocation is not clear and may not correct. Do you mean dry deposition or wet deposition vs. gas column concentrations? From Figure 9, the spatial distributions of dry deposition of oxidized and reduced nitrogen are rather consistent with their column concentrations. Also the discussion of conservations is not clear. Higher emissions would have higher depositions as both oxidized and reduced nitrogen have short lifetimes, and the differences between emissions and depositions do not reflect their concentrations in the air.

10. Page 14, Line 591-592:
"higher contribution of Nox to TIN in SE indicated more oxidant ratio of the precursors than NC". It is not clear what "more oxidant ratio of the precursors" means. Please clarify.

**Technical comments:** 1. Page 5, Line 194:
"US 25 National Aeronautics and Space Administration", should delete "25" here?

2. Page 8, Line 322:
"almost double higher than", should be "almost double"or "almost a factor of 2 higher than"

3. Page 12, Line 488:
What do you mean by "the correlated consistence"?

4. Page 12, Line 496:
"the magnitude difference", do you mean "large differences"?

5. Page 12, Line 510:
"this need to be" should be "this needs to be"

6. Page 13, Line 547:
What do you mean by "this conservation data"

7. Page 13, Line 555-558:
Use "major contributions" to denote "18

8. Page 14, Line 572:
Change "increasing trend" to "increasing order"

9. Page 14, Line 579:
Change "correspondingly" to "corresponding"

[Figure]

2020.

---

## Author Comment (AC1) · 14 Jun 2020

The authors appreciate the reviewers for reviewing our manuscript and providing constructive comments. As suggested, we carefully revised the manuscript thoroughly according to the valuable advices, as well as the typographical, grammatical, and bibliographical errors. Listed below are our point-by-point responses in blue to the review's comments (in italic).

**Anonymous Referee #2**

*This manuscript examines the performance of nine models within MICS-Asia III in capturing wet and dry deposition with observation from a plenty of sites. There are a number of grammar issues and typos. I would enough the authors to check carefully and improve the quality of writing. Besides, the scientific quality would be largely improved if the authors could provide more results to explain the differences between models and observations.*

**[Response]**: We would like to thank the reviewer for the valuable comments. Detailed explanation of differences of reactive nitrogen wet depositions between models and observations have been further discussed and been addressed as independent paragraphs in sections 3.1.1 and 3.1.2 of the revised manuscript.

"Sections 3.1.1:

*Regarding to the comparison over the whole of East Asia reported in the overview of acid deposition in MICS-Asia III (Itahashi et al., 2020),*

*similar overestimation was found in M5 and M11 while underestimation in M2, M4 and M14. It should be noted that the EANET sites are mostly located around Japan, Korea and Southeast Asia, and only 8 sites are located in China. The similar performances between the validation in East Asia and China indicated the general underestimation (overestimation) of M2, M4 and M14 (M5 and M11) were reliable in these models. For the rest of models, different results were found between China and East Asia, i.e., the simulated $N_{ox}$ wet deposition in M1 was significant overestimated in China (Figure 6 of Itahashi et al., 2020), but comparable with the observations over the rest of East Asia. Due to the absence of the observations for atmospheric $NO_2/NO_3^-$, we cannot validate their model performances directly. Instead, another companion paper (Chen et al., 2019b) reported that most of models overestimated $NO_3^-$ concentrations based on 14 sites in China with most sites located in NC (Figure S5 of Chen et al., 2019b). In summary, the relationship between the atmospheric concentration of $NO_3^-$ and the wet deposition in NC was not obvious, which is also same as that found in East Asia (Itahashi, et al., 2020).*"

"Sections 3.1.2:

*The underestimation of $N_{rd}$ wet deposition was also found over the whole of East Asia reported in the overview of acid deposition in MICS-Asia III*

*(Itahashi et al., 2020). This implies the current CTM models might underestimate prediction of $N_{rd}$ wet deposition not only in China but also in the whole of East Asia. The close correlations between the atmospheric concentration of $NH_4^+$ and wet deposition of $N_{rd}$ with overestimation in the atmosphere but underestimation in precipitation were found over all of East Asia (Itahashi et al., 2020). In this study, the consistent relationships in NC were also found in the results of Chen et al. (2019b) (overestimated $NH_4^+$ concentration) and in this study (underestimated $N_{rd}$ wet deposition). Bae et al. (2012) reported the below-cloud scavenging process was important in the simulation of $N_{rd}$ wet deposition, which was not explicitly separated in-cloud and below-cloud scavenging but computes it as a whole in the CMAQ model. Note that the wet scavenging process in most of models (including M11 and M12) of MICS-Asia III were similar with that treated in CMAQ module except M13 (Table 1). It is too simple to accurately simulate wet deposition with the absence of accurate below cloud wet scavenging simulation. This would be one reason for the underestimation of $N_{rd}$ wet deposition, especially considering the high concentration of gaseous ammonia in the surface layer of NC (Pan et al., 2018; Kong et al., 2019)."*

Besides, the grammar issues and typos have been corrected in the whole manuscript. After carefully check and polish of English, the quality of

writing has been improved according to the help of a native speaker. Following are the responses to the comments.

*Specific comments are listed below:*

*Line 52: Typo: WRF-CAMQ should be WRF-CMAQ*

**[Response]**: It has been revised.

*Line 105: Typo: shown should be showed*

**[Response]**: It has been revised.

*Line 123: uncompleted should be incomplete*

**[Response]**: It has been revised.

*Line 179: driving should be driven*

**[Response]**: It has been revised.

*Line 292: may not due to should be "may not be due to"*

**[Response]**: It has been revised.

*Lines 292-294: What would be the reason for the differences?*

*In many places, wrong tense is used. For example, in line 295: showed should be shows. Please also change the tense in other places.*

**[Response]**: It is clearly shown in Table 2 and Table 3 that the spatial correlation coefficients R is higher in urban sites than that in rural sites for $N_{ox}$ wet deposition and non-significant difference between the two categories sites for $N_{rd}$ wet deposition. There are many reasons including emissions, chemical conversions and deposition processes in the models might lead to the different performance of $N_{ox}$ wet deposition between urban and rural sites. However, the formations of $NO_3^-$ from its precursors $NO_x$ are more complicate than the other species due to the complicate chemical reactions. The companion paper (Chen et al., 2019) also showed the relatively low R value of the simulated $NO_3^-$ concentration at 31 sites over East Asia (0.29-0.65) compared with that of $SO_4^-$ (0.46-0.76), $NH_4^+$ (0.34-0.75), BC (0.65-0.80) and $PM_{2.5}$ (0.71-0.83) by 12 models in MICS-Asia III. This indicated that most of the current CTM models were more difficult to accurate predict complicated reactive species (i.e., $NO_3^-$) than inert substance (i.e., BC). Consider most of $NO_x$ were emitted in urban region, the more aged air mass that experienced complete degree of chemical reactions were usually characterized in rural area. Thus, the large difference in rural sites than that in urban sites among the multi-models would be reasonable. In response to your second comment, the tense in the whole manuscript has been checked and changed accordingly.

*Line 312-314: The differences might result from multiple reasons, including emissions, chemical conversions, deposition processes in the models, etc. There is no evidence or analysis showing that is caused by the coarse grid.*

**[Response]**: We agree with that the differences are from multiple reasons in the simulation of wet deposition process. We calculate the standard deviation (SD) of the observed and simulated yearly wet deposition for Nox in 9 sites over PRD region. Since these 9 sites are located relative close to each other (Figure 1 of manuscript), the SD represents the differences among the sites. The results showed that the observed SD was 5.85 and was much higher than all of the simulated SD (0.90-3.12). This indicated the coarse grid couldn't capture the large differences in the observations at a local scale.

*Lines 335-343: This could be one of the reasons with the assumption that models can accurately capture these processes. Is it possible to compare rainfall events, which does not require high resolution of deposition data*

**[Response]**: Thank you for this valuable comment. We agree that the detailed validation such as the duration and intensity of the rainfall events could be helpful to better understand the performance of the wet scavenging process in the model. However, the sampling interval of wet deposition is mostly daily, weekly, or event yearly in Southeast of China

and North of China. Additionally, the sampling periods of these measurements were not consistent across site. Besides, the current meteorological models have difficulty in capturing the timing of precipitation events. We believe the comparison between observations and simulations in monthly data of wet deposition is an appropriate approach and this analysis could provide a broad overview in China currently. Anyway, the rainfall event simulation is our future target and will be taken more focus in the next.

*Lines 575-576: correlations between observed depositions between emissions*

**[Response]**: It has been revised.

*Line 662: importance should be important*

**[Response]**: It has been revised.

Reference:

Bae, S. Y., Park, R. J., Yong, P. K., and Woo, J. H.: Effects of below-cloud scavenging on the regional aerosol budget in East Asia, Atmos Environ, 58, p.14-22, 2012.

Itahashi, S., Ge, B., Sato, K., Fu, J. S., Wang, X., Yamaji, K., Nagashima, T., Li, J., Kajino, M., Liao, H., Zhang, M., Wang, Z., Li, M., Kurokawa, J., Carmichael, G. R., and Wang, Z.: MICS-Asia III: Overview of model inter-comparison and evaluation of acid deposition over Asia, Atmos. Chem. Phys., 2019, 1-53, 10.5194/ acp-20-2667-2020, 2020.

Kong, L., Tang, X., Zhu, J., Wang, Z., Pan, Y., Wu, H., Wu, L., Wu, Q., He, Y., Tian, S., Xie, Y., Liu, Z., Sui, W., Han, L., and Carmichael, G.: Improved Inversion of Monthly Ammonia Emissions in China Based on the Chinese Ammonia Monitoring Network and Ensemble Kalman Filter, Environ Sci Technol, 53,

12529-12538, 10.1021/acs.est.9b02701, 2019.

Pan, Y., Tian, S., Zhao, Y., Zhang, L., Zhu, X., Gao, J., Huang, W., Zhou, Y., Song, Y., and Zhang, Q.: Identifying ammonia hotspots in China using a national observation network, Environ Sci Technol, 2018.

Tan, J., Fu, J. S., Carmichael, G. R., Itahashi, S., Tao, Z., Huang, K., Dong, X., Yamaji, K., Nagashima, T., Wang, X., Liu, Y., Lee, H. J., Lin, C. Y., Ge, B., Kajino, M., Zhu, J., Zhang, M., Hong, L., and Wang, Z.: Why models perform differently on particulate matter over East Asia? – A multi-model intercomparison study for MICS-Asia III, Atmos. Chem. Phys. Discuss., 2019, 1-36, 10.5194/acp-2019-392, 2019.

---

## Author Comment (AC2) · 14 Jun 2020

The authors appreciate the reviewers for reviewing our manuscript and providing constructive comments. As suggested, we carefully revised the manuscript thoroughly according to the valuable advices, as well as the typographical, grammatical, and bibliographical errors. Listed below are our point-by-point responses in blue to the review's comments (in italic).

**Anonymous Referee #1**

*This manuscript has presented the analyses of atmospheric nitrogen deposition to China as simulated by an ensemble of chemical transport models participating the MICS-Asia III model intercomparison. Available surface measurements of wet deposition fluxes are integrated to assess the model performances. This represents an important step towards a better understanding the model range and uncertainties in simulating nitrogen deposition. Different from previous studies on multi-model nitrogen deposition simulation, most models analyzed in this study used the same emissions and driving meteorology, allowing a closer attribution of the factors driving the model uncertainties. The results show that most models calculated consistent spatial and temporal variations of nitrogen deposition for both oxidized and reduced nitrogen, yet considerable differences exist among models.*

*I think the study is an important contribution to the MICS-Asia III special issue. The analyses are mostly fine, and it would be much scientifically stronger if having a deeper investigation on the drivers of model*

*differences. The following comments also need to be addressed. In addition to my specific comments as elaborated below, improvements on the language are necessary and need caution.*

**[Response]**: We would like to thank the reviewer for the valuable comments. Deeper discussion on the difference among the models as well as the difference between models and observations have been added in the revised manuscript. We also invite a native speaker to polish the language, which is mentioned in the acknowledgment. All the revision in the manuscript has been marked in blue. Following are the responses to the comments.

**Specific comments:**

*1. Page 5, Line 199-202: How about natural sources of nitrogen, e.g., nitrogen oxides from soil and lightning? Are they included in any of these models?*

**[Response]:** No, the nitrogen oxides emission from soil and lightning are not included in MICS-Asia III. The natural sources included in this project are biogenic emissions from MEGANv2.4 and biomass burning emissions from GFEDv3.

*2. Page 6, Line 220-223: Did any of the models also simulate dry deposition of other nitrogen species, e.g., PAN, isoprene nitrates? How*

*important are these nitrogen species contributing to dry deposition, and the uncertainty induced by excluding them in the analysis? Please clarify.*

**[Response]:** Unfortunately, the PAN as well as isoprene nitrates are not included in the simulated dry deposition in this study. PAN is an important photochemical product formed from the reactions between VOCs and NOx under sunlight. The loss of PAN was modulated mainly by dry deposition and horizontal transport (Yuan et al., 2018). However, their contributions to total N dry deposition are less important than the inorganic N, e.g., $HNO_3$ and $NO_x$. A comparison between the Chemistry of Atmosphere-Forest Exchange (CAFE) Model and BEARPEX-2007 observations in California has been implemented and found that the $HNO_3$ dominate total dry deposition of oxidized N (~83%) in warm seasons, which indicated the other $NO_y$ (including NOx, PANs, etc) may take up less than 17%. (Wolfe1 et al., 2011). Besides, several studied also investigated the Organic N (ON) deposition accounted for about 20–30% of total N compounds in wet and dry deposition (Duce et al., 2008; Benitez et al., 2009). Thus, the uncertainties of excluding other nitrogen species, e.g., PAN, isoprene nitrates from the total dry deposition could be negligible compared with the uncertainties of the simulated $HNO_3$ and $NH_3$ dry deposition in this study. We have added this interpretation in our revised manuscript.

*3. Page 7, Line 261: It is not clear how the data are normalized as monthly wet deposition fluxes. Do you mean that the raw measurements are at different temporal resolutions (daily, weekly, etc.), and then are interpolated to monthly values? This shall be explained in the text.*

**[Response]:** Yes. Since the measured wet deposition data are collected from different sources, the temporal resolutions of the data are different from each other, i.e., daily in EANET and CREN, rainfall event collection in DEE and yearly in NNDMN, which has been mentioned in sec.2.3 L237-259 in the original manuscript. We have added the explanation in the revised manuscript, as " *The temporal resolutions of the wet deposition data are different from each other, i.e., daily in EANET and CREN, rainfall event collection in DEE and yearly in NNDMN. For the convenience of comparison, all data from daily or rainfall event collecting samples at each type of measurement site has been summarized and interpolated as monthly wet deposition data to compare with the monthly simulation in MICS-Asia III in this study, except the yearly data provided by NNDMN*".

*4. Page 7, Line 279-281: This sentence is not clear. Do you mean the correlation coefficients are lower than the value when only comparing with EANET data? Please clarify.*

**[Response]:** Yes. The EANET data used in this study is only 8 sites located in China (Supplementary material), which is different from the

whole EANET sites over East Asia listed in our companion paper (Itahashi et al., 2020). To avoid misunderstanding, the EANET sites used in this study are clarified as EANET sites in China in the revised Supplementary material. Meanwhile, the context in first paragraph of Section 3.1.1 has been revised as "*The NME was around 50% with the highest 82.2%, in M11, which is lower than that reported over the East Asia with the value of 70% based on EANET observation by Itahashi et al. (2020). However, the correlation coefficient R was around 0.2~0.3 (n=83) which is lower than the East Asia comparison based on the EANET data (0.3~0.4, n=54) (Itahashi et al., 2020)*".

*5. Page 7, Line 285-286: Need to add a sentence defining FAC2.*

**[Response]:** It has been added in the revised manuscript as "*To judge the agreement between simulation and observation, the percentages within a factor of 2 (FAC2) has been calculated in this study.*"

*6. Page 10, Line 417-423: The sentence here needs rewritten or removed. It is a long sentence, and the information is repetitive in the paragraph.*

**[Response]:** Agree. The sentence here has been removed in the revised manuscript.

*7. Page 12, Line 478-480: Is Figure 8 for the surface layer or the atmospheric column? As dry deposition only applies to species at the surface layer, while wet deposition can extend to the whole tropospheric*

*column, an explanation is needed here to justify why you use it for both dry and wet depositions.*

**[Response]:** Figure 8 is the CV of the related species' concentration in surface layer. It is indeed that the wet deposition can extend to the whole column through in-cloud and below-cloud scavenging process. However, the simulated data for most of the air concentrations in MICS-Asia III were at the surface layer except $NO_2$, which also included the vertical column density (VCD) data. As for $NO_2$ VCD, the CV results show that the similar spatial distribution with that in surface layer (Figure S9 in revised supporting material). This indicated that the simulated concentration at surface layer could partly represent the differences of the whole column among the multi-models, especially in providing a broad overview in China. In this study, the CV of the related air mass concentrations at surface layer has been calculated and compared with that both in dry and wet deposition to explain the reasons for the differences among the simulated depositions in MICS-Asia III. The explanation has been added in the revised version. "*It should be noted that only concentrations of most of the related species at surface layer have been submitted in MICS-Asia III, except $NO_2$ vertical column density data (VCD). According to the comparison of CV between the $NO_2$ concentration at the surface layer and VCD data (Figure S5), it was shown that there is a similar spatial pattern over the whole of China. This*

*indicates that the simulated concentration on the surface layer can partly represent the difference of the whole column among the multi-models, especially in providing a broad overview in China. Thus, only concentration data at the surface layer has been used in this study.*"

*8. Page 12, Line 500-510: How about dry deposition velocities? Did all the models calculate the dry deposition fluxes as the products of surface concentration and dry deposition velocity? It is missing something that the discussion of dry deposition only examined concentrations and not include dry deposition velocities.*

**[Response]:** Thank you for suggestion. We have conducted the analysis of dry deposition velocities based on the ratio of the dry deposition fluxes and the surface concentration simulated by each model and prepared additional Figure S6~Figure S11 in the revised supporting material. These points have now been addressed in the last paragraph in Section 3.3.2.

"*For $N_{ox}$ dry depositions, the anomalies of deposition and $NO_x$ concentration in the air are shown in Figure S6 and Figure S7. It shows same variations among the models, i.e., both of higher deposition and concentration in M1, M5, M11, M13, and lower in M2, M4, M6, M12 and M14. This has also been proved in the distribution of CV values in $NO_x$ (Figure 8a) and $N_{ox}$ dry depositions (Figure 7a) during autumn and winter. As the most important precursor of $N_{rd}$ dry deposition, gaseous*

*NH₃ also shows large CV values in central China during summer time (> 0.6). There were also significant high CV values in south of the Yangtze River during the autumn and winter period (0.7-0.8 in south of the Yangtze River vs 0.3-0.5 in north of the Yangtze River). A similar pattern but of uncertain significance was found in the simulated $N_{rd}$ dry deposition (0.3-0.4 vs 0.2-0.3 in Figure 7b). The anomalies of $N_{rd}$ dry deposition and the gaseous NH₃ in the air simulated by each model are shown in Figure S8 and Figure S9. Addintionally, the dry deposition velocity ($V_d$) of $N_{rd}$ - based on the ratio of the dry deposition fluxes and the surface concentration (same as Tan et al., 2019) - are also shown in Figure S10. The results show that the CMAQ models (M1~M6) predicted similar $V_d$ of $N_{rd}$, and the $N_{rd}$ dry deposition was consistent with the gaseous NH₃ concentration in the air, i.e., overestimation in M1 and M2 but underestimation in M4 and M5. However, among the different models, high $V_d$ of $N_{rd}$ (low $V_d$ of $N_{rd}$) was corresponds with the overestimation (underestimation) of dry deposition in M11 and M14 (M12 and M13). From the distribution of CV, similar patterns were also displayed both in $V_d$ (Figure S11) and dry deposition of $N_{rd}$, with low CV value in NCP (0.1-0.4 for $N_{rd}$ dry deposition, 0.1-0.3 for $V_d$) and high CV value in SE and SW (0.4-0.8 for $N_{rd}$ dry deposition, higher than 0.5 for $V_d$)."*

*9. Page 13, Line 538-548: The discussion of different allocation is not clear and may not correct. Do you mean dry deposition or wet deposition*

*vs. gas column concentrations? From Figure 9, the spatial distributions of dry deposition of oxidized and reduced nitrogen are rather consistent with their column concentrations. Also the discussion of conservations is not clear. Higher emissions would have higher depositions as both oxidized and reduced nitrogen have short lifetimes, and the differences between emissions and depositions do not reflect their concentrations in the air.*

**[Response]:** We agree with that the differences between emissions and depositions do not reflect their concentrations in the air. The chemical transformation as well as the regional transport may also affect their atmospheric concentration in the air. The purpose of this discussion is to validate the reasonable distribution of the simulated depositions through the comparison between $N_{ox}$ and $N_{rd}$, instead of the difference between deposition and gas column concentrations. To address the issues in your comment, the last paragraph of section 4.1 has been revised as:

"*Another interesting phenomenon is that the allocations of high values of depositions and VCD of $N_{ox}$ are different from that of $N_{rd}$. As shown in the Figure 9, low depositions with high values of VCD for $N_{ox}$ and high depositions with comparatively lower level of VCD for $N_{rd}$ co-existed in East China. On a global scale, air pollutants must follow the conservation law - that is, the emissions can be divided into two parts, i.e., the depositions and their concentrations in the air. Here we apply this*

*concept to the entire region of China. We assume that the amount of $N_{ox}$ and $N_{rd}$ transported out of the research areas is equivalent under the same atmospheric advection. The emissions of $N_{ox}$ and $N_{rd}$ in China are also comparable (8.3 kg $N \bullet ha^{-1}$ and 8.7 kg $N \bullet ha^{-1}$ for $NO_x$ and $NH_3$, respectively). At the same time, the simulated low deposition of $N_{ox}$ and observed high VCD match exactly with the high deposition in $N_{rd}$ and observed low VCD in central and eastern China. Although there is no directly observed distribution map to verify the total deposition in our models, the close correlation between the observed VCD and deposition can verify the rationality of the simulated total deposition distribution."*

10. *Page 14, Line 591-592: "higher contribution of Nox to TIN in SE indicated more oxidant ratio of the precursors than NC". It is not clear what "more oxidant ratio of the precursors" means. Please clarify.*

**[Response]:** The "more oxidant ratio of the precursors" means higher nitrogen oxidant ratio (i.e., the ratio of oxidation from $NO_2$ to $NO_3^-$). According to your comment, the sentence has been revised as *"Considering the lower ratio of $NO_x/NH_3$ emission in SE (21.4/21.6, 0.99) than NC (30.4/24.4, 1.25), higher contribution of $N_{ox}$ to TIN in SE indicated a higher nitrogen oxidant ratio (i.e., the ratio of oxidation from $NO_2$ to $NO_3^-$) than NC. Our companion paper (Tan et al., 2019) also revealed the higher nitrogen oxidation ratio in SE as 0.4-0.6, compared with that in NC as 0.2-0.4."*

**Technical comments:**

*1. Page 5, Line 194: "US 25 National Aeronautics and Space Administration", should delete "25" here?*

**[Response]:** It has been deleted in the revised manuscript.

*2. Page 8, Line 322: "almost double higher than", should be "almost double"or "almost a factor of 2 higher than"*

**[Response]:** It has been replaced by "*almost double*" in the revised manuscript.

*3. Page 12, Line 488: What do you mean by "the correlated consistence"?*

**[Response]:** The correlated consistence here means the similar variation of CV in the simulated particulate $NO_3^-$ concentration in the air mass and $N_{ox}$ wet deposition were shown in Figure 7 (c) and 8 (a). To avoid misunderstanding, the "*correlated consistence*" has been replaced as "*consistent distribution of CV*" in the revised manuscript.

*4. Page 12, Line 496: "the magnitude difference", do you mean "large differences"?*

**[Response]:** Yes, the difference can reach at magnitude level. It has been replaced by "*large difference even at magnitude level*" in the revised manuscript.

*5. Page 12, Line 510: "this need to be" should be "this needs to be"*

**[Response]:** It has been changed in the revised manuscript.

*6. Page 13, Line 547: What do you mean by "this conservation data"*

**[Response]:** The "conservation data" means that the simulated low deposition in $N_{ox}$ and observed high VCD match exactly with the high deposition in $N_{rd}$ and observed low VCD in central and eastern China. This phenomenon was perfectly constrained by the comparable emissions between $N_{ox}$ and $N_{rd}$ in China. In avoid to misunderstanding, these expressions have been deleted. Detailed information could be found in the response to specific comment 9.

*7. Page 13, Line 555-558: Use "major contributions" to denote "18*

**[Response]:** The "*major contributions*" have been revised as "*important contributions*" in the revised manuscript.

*8. Page 14, Line 572: Change "increasing trend" to "increasing order"*

**[Response]:** It has been changed accordingly.

*9. Page 14, Line 579: Change "correspondingly" to "corresponding"*

**[Response]:** It has been changed accordingly.

Reference:

Benitez, J. M. G., Cape, J. N., Heal, M. R., van Dijk, N., and Diez, A. V.: Atmospheric nitrogen deposition in south-east Scotland: Quantification of the organic nitrogen fraction in wet, dry and bulk deposition, Atmos Environ, 43, 4087-4094, 10.1016/j.atmosenv.2009.04.061, 2009.

Duce, R. A., LaRoche, J., Altieri, K., Arrigo, K. R., Baker, A. R., Capone, D. G., Cornell, S., Dentener, F., Galloway, J., Ganeshram, R. S., Geider, R. J., Jickells, T., Kuypers, M. M., Langlois, R., Liss, P. S., Liu, S. M., Middelburg, J. J., Moore, C. M., Nickovic, S., Oschlies, A., Pedersen, T., Prospero, J., Schlitzer, R., Seitzinger, S., Sorensen, L. L., Uematsu, M., Ulloa, O., Voss, M., Ward, B., and Zamora, L.: Impacts of atmospheric anthropogenic nitrogen on the open ocean, Science, 320, 893-897, 10.1126/science.1150369, 2008.

Itahashi, S., Ge, B., Sato, K., Fu, J. S., Wang, X., Yamaji, K., Nagashima, T., Li, J., Kajino, M., Liao, H., Zhang, M., Wang, Z., Li, M., Kurokawa, J., Carmichael, G. R., and Wang, Z.: MICS-Asia III: Overview of model inter-comparison and evaluation of acid deposition over Asia, Atmos. Chem. Phys., 2019, 1-53, 10.5194/ acp-20-2667-2020, 2020.

Tan, J., Fu, J. S., Carmichael, G. R., Itahashi, S., Tao, Z., Huang, K., Dong, X., Yamaji, K., Nagashima, T., Wang, X., Liu, Y., Lee, H. J., Lin, C. Y., Ge, B., Kajino, M., Zhu, J., Zhang, M., Hong, L., and Wang, Z.: Why models perform differently on particulate matter over East Asia? – A multi-model intercomparison study for MICS-Asia III, Atmos. Chem. Phys. Discuss., 2019, 1-36, 10.5194/acp-2019-392, 2019.

Wolfe, G. M., Thornton, J. A., Bouvier-Brown, N. C., Goldstein, A. H., Park, J. H., McKay, M., Matross, D. M., Mao, J., Brune, W. H., LaFranchi, B. W., Browne, E. C., Min, K. E., Wooldridge, P. J., Cohen, R. C., Crounse, J. D., Faloona, I. C., Gilman, J. B., Kuster, W. C., de Gouw, J. A., Huisman, A., and Keutsch, F. N.: The Chemistry of Atmosphere-Forest Exchange (CAFE) Model - Part 2: Application to BEARPEX-2007 observations, Atmos Chem Phys, 11, 1269-1294, 10.5194/acp-11-1269-2011, 2011.

Yuan, J., Ling, Z., Wang, Z., Lu, X., Fan, S., He, Z., Guo, H., Wang, X., and Wang, N.: PAN-Precursor Relationship and Process Analysis of PAN Variations in the Pearl River Delta Region, Atmosphere-Basel, 9, 10.3390/atmos9100372, 2018.

---

## Author Response (AR2)

The authors appreciate the reviewers for reviewing our manuscript and providing constructive comments. As suggested, we carefully revised the manuscript thoroughly according to the valuable advices, as well as the technical errors. Listed below are our point-by-point responses in blue to the review's comments.

The revised manuscript has some large improvements and has fairly addressed the reviewers' comments. I have two remaining comments that I think the manuscript shall clarify before it can be accepted on ACP.

(1) Page 10, Sect. 3.1.2, Line 383-400 -

The study attributed to the model underestimates of Nrd wet deposition to the parameterizations of wet scavenging. Are the same wet scavenging parameterizations apply to Nox wet deposition? If so, the bias in parameterization shall also affect the Nox wet deposition simulation. How about uncertainty in the NH3 emissions? Would that contribute to the underestimates in Nrd wet deposition simulations? Please clarify.

[Response]: Yes. The wet scavenging parameterizations should also be applied to $N_{ox}$ wet deposition, which also lead to some underestimation of $N_{ox}$ wet deposition in SE and SW+TP where is referred as the higher load of precipitation area in China. However, the affect factors of $N_{ox}$ wet deposition are more complicate than $N_{rd}$ wet deposition due to their chemical reactivity. This therefore leads to larger uncertainties in the

simulation of both air concentration and deposition of $N_{ox}$ in MICS-Asia III.

The uncertainty in the $NH_3$ emissions is also an important factor for the underestimation of $N_{rd}$ wet deposition in China. Kong et al. (2019) inversed a monthly $NH_3$ emission in China based on the Chinese Ammonia Monitoring Network and Ensemble Kalman Filter. They found a significant underestimation of $NH_3$ emission especially in NCP. This has been added in the revised manuscript as *"Besides, the underestimation $NH_3$ emission in China would also lead to the underestimation of $N_{rd}$ wet deposition. According to the improved inversion of $NH_3$ emission in China by Kong et al. (2019), the significant underestimation of $NH_3$ emission was found especially in NCP. For the whole China, the priori emission and the inversion emission of $NH_3$ are 10.3 Tg/year and 13.1 Tg/year, respectively."*

(2) Page 14, Sect. 4.1, Line 592-606

I still did not get from Figure 9 what "the allocations of high values of depositions and VCD of Nox are different from that of Nrd" means. Were you discussing dry deposition or wet deposition? As seen from Figure 9, the patterns of dry deposition for both Nox and Nrd are consistent with their VCD patterns, while some differences exist for wet deposition, which are very likely driven by precipitation.

The statement "the emissions can be divided into two parts, i.e., the

depositions and their concentrations in the air" also did not make sense to me. This is not an equation of conservation. Emissions and depositions are fluxes with unit of mass per unit time, while concentrations in air reflect total mass.

**[Response]:** No. The deposition here is discussed as a total instead of each part. The different allocation of high values of depositions and VCD for $N_{ox}$ and $N_{rd}$ means the relative higher value in VCD $NO_2$ and deposition $N_{rd}$, while relative lower value in deposition $N_{ox}$ and VCD $NH_3$ in central and eastern China. Take the similar emission for both $NO_x$ and $NH_3$ into account, the close negative correlation between the observed VCD and deposition for the two type of nitrogen, e.g., $N_{ox}$ and $N_{rd}$, means the rationality of the simulated total deposition distribution. However, it is true that the deposition and concentration in the air cannot conserve to the total emission due to the different unit. To avoid the misunderstanding, the paragraph was deleted in the revised version.

Technical comments:

Page 6, Line 224: There are two Wolfe et al. 2011, and Wolfe1 shall be Wolfe

**[Response]:** Yes. The reference has been revised accordingly.

Figure 5 and 6, the text and numbers in panel (l) are too small to read.

**[Response]:** The text in panel (l) of Figure 5 and Figure 6 has been enlarged.